# Piezo1 channels sense whole body physical activity to reset cardiovascular homeostasis and enhance performance

Baptiste Rode[1], Jian Shi[1], Naima Endesh[1], Mark J. Drinkhill[1], Peter J. Webster[1], Sabine J. Lotteau[2], Marc A. Bailey[1], Nadira Y. Yuldasheva[1], Melanie J. Ludlow[1], Richard M. Cubbon[1], Jing Li[1], T.Simon Futers[1], Lara Morley[1], Hannah J. Gaunt[1], Katarzyna Marszalek[1], Hema Viswambharan[1], Kevin Cuthbertson[3], Paul D. Baxter[1], Richard Foster[3], Piruthivi Sukumar[1], Andrew Weightman[4], Sarah C. Calaghan [2], Stephen B. Wheatcroft[1], Mark T. Kearney[1] & David J. Beech[1]

Mammalian biology adapts to physical activity but the molecular mechanisms sensing the activity remain enigmatic. Recent studies have revealed how Piezo1 protein senses mechanical force to enable vascular development. Here, we address Piezo1 in adult endothelium, the major control site in physical activity. Mice without endothelial Piezo1 lack obvious phenotype but close inspection reveals a specific effect on endothelium-dependent relaxation in mesenteric resistance artery. Strikingly, the Piezo1 is required for elevated blood pressure during whole body physical activity but not blood pressure during inactivity. Piezo1 is responsible for flow-sensitive non-inactivating non-selective cationic channels which depolarize the membrane potential. As fluid flow increases, depolarization increases to activate voltage-gated $Ca^{2+}$ channels in the adjacent vascular smooth muscle cells, causing vasoconstriction. Physical performance is compromised in mice which lack endothelial Piezo1 and there is weight loss after sustained activity. The data suggest that Piezo1 channels sense physical activity to advantageously reset vascular control.

[1] Schools of Medicine, University of Leeds, Leeds LS2 9JT, UK. [2] School of Biomedical Sciences, University of Leeds, Leeds LS2 9JT, UK. [3] School of Chemistry, University of Leeds, Leeds LS2 9JT, UK. [4] School of Mechanical, Aerospace and Civil Engineering, University of Manchester, Manchester M13 9PL, UK. Baptiste Rode, Jian Shi and Naima Endesh contributed equally to this work. Correspondence and requests for materials should be addressed to D.J.B. (email: d.j.beech@leeds.ac.uk)

The health value of exercise has been described since the time of Hippocrates, but it was not until 1953 that it was demonstrated scientifically[1]. It is now known that whole body physical activity and other forms of physical exercise afford major protection against chronic disease[2, 3]. Such protection seems likely to have evolved because animals survived by tuning their biology to regular physical activity to avoid predators and source prey and other food. Without this activity many humans today are in suboptimal environments, increasing the risk of dysregulation and disease. Therefore there has been intense research into exercise[2]. However, the existence and identity of molecular sensors of exercise has remained unclear. If we could identify such sensors, we might find ways to better tune human biology to advanced societies. Because the likely site of exercise sensors is the endothelium[2], we were interested in whether Piezo1 might act as an exercise sensor. Piezo1 is a relatively recently discovered membrane protein which assembles as a trimer to form $Ca^{2+}$-permeable non-selective cationic channels activated by physical force[4–6]. It is highly expressed in endothelial cells and known to be important for integrating vascular architecture with physical force during embryonic development[7, 8].

In this study, we conditionally disrupted Piezo1 in the endothelium to investigate its relevance to adult mice. We found that elevated blood pressure of whole body physical exercise depended on endothelial Piezo1. The mechanism was a vascular bed-specific effect of Piezo1 which opposed endothelium-dependent relaxation mediated by endothelium-derived hyperpolarization (factor) (EDH(F)) to cause vasoconstriction when fluid flow was elevated. We conclude that endothelial Piezo1 is an exercise sensor which enables optimized redistribution of blood flow to enhance physical performance.

## Results

**Mice with disrupted endothelial Piezo1 are superficially normal.** To investigate the relevance of Piezo1 in the adult endothelium, we engineered mice with conditional Cre-Lox-mediated disruption of Piezo1 in the endothelium (Piezo1$^{\Delta EC}$ mice) (Supplementary Figs. 1 and 2). The mice appeared normal and had normal body weights, weight gains and organ weights and serum urea, $K^+$ and $Na^+$; gross anatomies and functions of the heart and aorta were also normal (Fig. 1a–m) (Supplementary Fig. 3). Retinal vasculature and endothelial response to injury were normal (Fig. 1n–q). Therefore endothelial Piezo1 in the adult appeared to be without consequence.

To investigate the mice in more detail, we made isometric tension recordings from second-order mesenteric arteries, looking for relevance of Piezo1 to endothelium-dependent tone. As expected, arteries from control genotype mice contracted in response to the $\alpha_1$-adrenoceptor agonist phenylephrine and then relaxed in response to the endothelium-dependent vasodilator acetylcholine (Fig. 2a, b). Arteries from Piezo1$^{\Delta EC}$ mice behaved similarly (Fig. 2c, d). These data also suggested that endothelial Piezo1 in the adult was of no consequence.

**Endothelial Piezo1 channels have an anti-EDH(F) effect.** The $Ca^{2+}$-permeable non-selective cationic pathway generated by Piezo1 channels[4–6] poses an intriguing dichotomy for endothelial biology (Fig. 2e). $Ca^{2+}$ entry theoretically drives $Ca^{2+}$ dependent mechanisms such as activation of endothelial nitric oxide synthase[7, 9]. But cation entry as a whole (and especially entry of the monovalent ion $Na^+$) theoretically drives membrane potential depolarization which could be important as a vasoconstrictor mechanism because EDH(F) is well established as a mechanism for endothelium-dependent vasodilatation[10]. Therefore, we investigated the effect of inhibiting EDH(F) by the established

method of combining apamin and charybdotoxin; toxin inhibitors of two $K^+$ channels which are critical for EDH(F)[10]. In mesenteric arteries from control genotype mice the toxins caused slight inhibition of the ACh response but they strongly inhibited the ACh response in arteries from Piezo1$^{\Delta EC}$ mice (Fig. 2f–i). Other properties of the arteries were unchanged by Piezo1$^{\Delta EC}$ (Supplementary Figs. 4, 5). The data suggest that endothelial Piezo1 channels oppose EDH(F) and may have specific roles under certain circumstances.

**Importance for elevated blood pressure in exercise.** Because of the relevance of mesenteric arteries to peripheral resistance and thus blood pressure, we inserted telemetry probes for continuous recording of blood pressure. Mice were provided with free access to a running wheel. Blood pressure was not different in Piezo1$^{\Delta EC}$ mice during periods of physical inactivity (Fig. 3a). In contrast, the increase in blood pressure seen during physical activity was reduced (Fig. 3b). The data suggest that endothelial Piezo1 has specific importance in blood pressure regulation during whole body physical activity.

**Constitutive and flow-induced Piezo1 in mesenteric artery.** To investigate how Piezo1 might mediate an anti-EDH(F) effect and elevation of blood pressure, we first developed a unidirectional monovalent cation flux assay for Piezo1 channels which showed strong monovalent cation permeability when the channels were over-expressed in HEK293 cells or natively expressed in cultured endothelial cells (Supplementary Fig. 6). Constitutive monovalent cation flux was evident, suggesting non-inactivating Piezo1 channels capable of mediating sustained depolarization (Supplementary Fig. 6).

To determine the relevance to physiological cells, we acutely isolated endothelial cells from second-order mesenteric arteries. Cell-detached outside-out membrane patches were used to enable identification of Piezo1 channels by their unitary current size (and therefore unitary conductance) and avoid contaminating effects from the cytosol and intracellular organelles. Outside-out patches are outwardly convex[11] and so the patch pipette did not protect the membrane from fluid flow. Dominant constitutive channel activity was observed with the expected 25 pS unitary conductance of Piezo1 channels (Fig. 4a, b)[4, 5]. Fluid flow enhanced the activity (Fig. 4a, c). Piezo1 channel identity was confirmed by sensitivity to inhibition by $Gd^{3+}$ (Fig. 4a, c) which blocks Piezo1 channels[4], and absence of the channels in patches from Piezo1$^{\Delta EC}$ mice (Fig. 4c, d). There was no response to fluid flow in the absence of Piezo1 (Fig. 4c, d). Moreover the fluid flow effect was mimicked by Yoda1 (Supplementary Fig. 7), a small-molecule activator of Piezo1 channels[12, 13]. The data suggest that constitutive and fluid flow-enhanced Piezo1 channel activity is common in mesenteric artery endothelial cells and that these channels do not inactivate or depend on intracellular factors.

To explore the significance in membrane potential control, freshly isolated sheets of mesenteric endothelium were used for measurements without contamination from other cell types. The sheets were syncitia of endothelial cells[14] to which a patch-clamp pipette was attached in whole-cell recording mode. The resting membrane potential averaged about −46 mV in static resting conditions and application of fluid flow caused reversible depolarization (Fig. 5a, b). Importantly, application of the Piezo1 channel inhibitor $Gd^{3+}$ or deletion of Piezo1 (Piezo1$^{\Delta EC}$) caused hyperpolarization of the resting potential to an average of about −60 mV, and fluid flow and Yoda1 now had no depolarizing effect (Fig. 5a, b) (Supplementary Fig. 7). Incremental increases in fluid flow up to and above fluid flow rates reported in

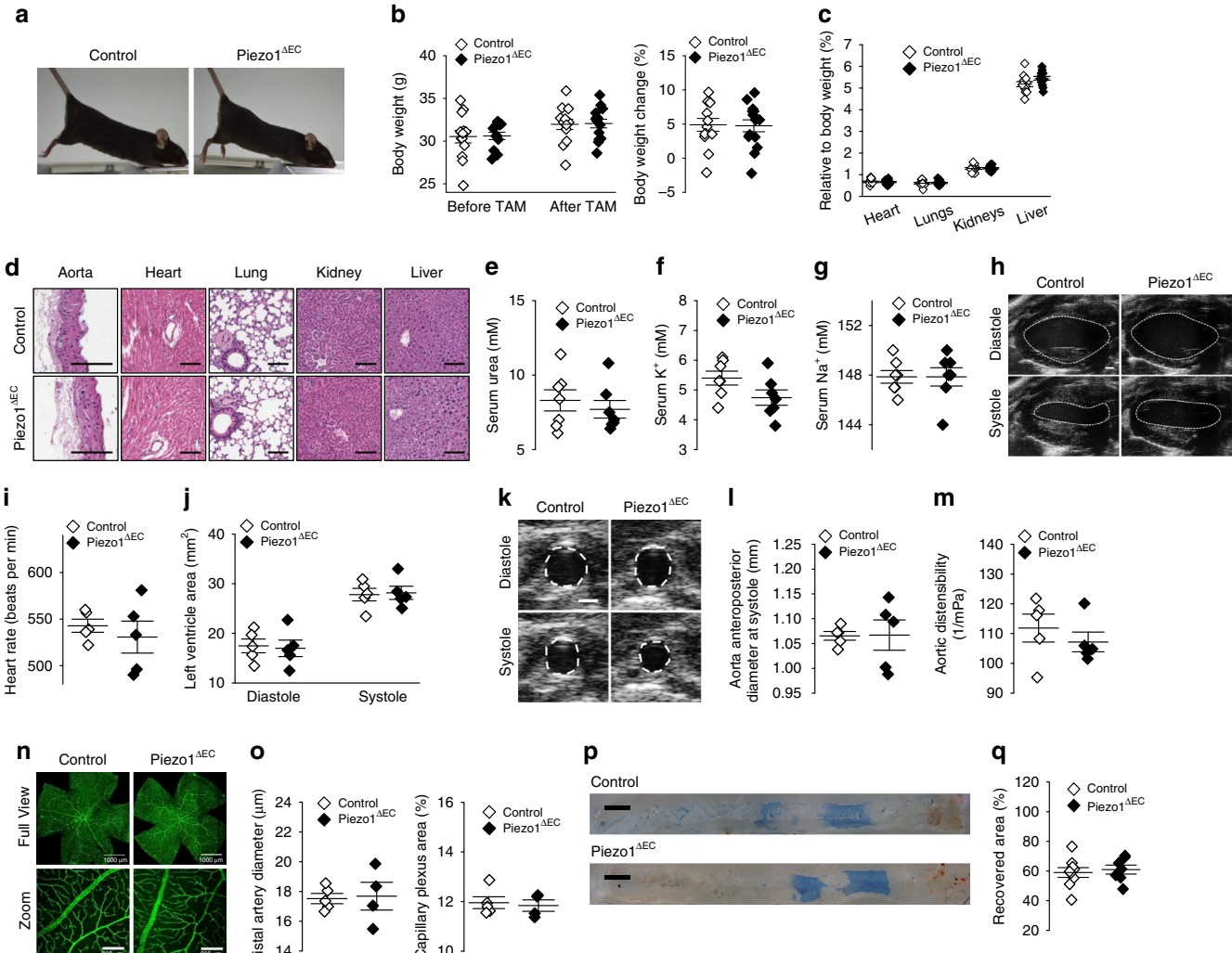

**Fig. 1** Mice with disrupted endothelial Piezo1 are superficially normal. **a** Physical appearance of control and endothelial Piezo1-deleted (Piezo1$^{\Delta EC}$) mice. **b** Body weight and percentage change in body weight of control ($n = 13$) and Piezo1$^{\Delta EC}$ ($n = 14$) mice before and 10–14 days after tamoxifen (TAM) treatment. **c** As percentages of total body weight, weights of heart, lung, kidney and liver in control ($n = 13$) and Piezo1$^{\Delta EC}$ ($n = 14$) mice. **d** Histological examples of control (*top row*) and Piezo1$^{\Delta EC}$ (*bottom row*) sections of aorta, heart, lung, kidney and liver stained with H&E. *Scale bars* 100 μm. **e**–**g** Serum concentrations of urea, K$^+$ and Na$^+$ in control ($n = 7$) and Piezo1$^{\Delta EC}$ ($n = 7$) mice. **h**–**m** Ultrasound study of the heart **h**–**j** and aorta **k**–**m** of control ($n = 5$) and Piezo1$^{\Delta EC}$ ($n = 5$) mice under anaesthesia. **h** Example of left ventricle images of control and Piezo1$^{\Delta EC}$ at diastole and systole. The left ventricle chamber is *circled* with a *white dashed line*. *Scale bar* 1 mm. **i**, **j** Cardiac parameters measured by ultrasound. **k** Example of aorta images of control and Piezo1$^{\Delta EC}$ at diastole and systole. The left ventricle chamber is circled with a *white dashed line*. *Scale bar* 1 mm. **l** Aorta anteroposterior diameter at systole. **m** Aortic distensibility. **n** Retinal vasculatures stained with isolectin (*green*) from control and Piezo1$^{\Delta EC}$ mice. Entire retinas (*full view*) and close up views (*zoom*). **o** Quantification of retina distal artery diameter and capillary plexus area from control ($n = 5$) and Piezo1$^{\Delta EC}$ ($n = 4$) mice. **p** Endothelial regeneration 5 days after femoral artery injury. Images of the arteries in which the *blue colour* shows Evans blue staining of areas which were not re-endothelialized after injury. *Scale bars* 0.5 mm. **q** Quantification of endothelial regeneration in control ($n = 9$) and Piezo1$^{\Delta EC}$ ($n = 7$) mice. Independent data points are displayed with superimposed *bars* indicating mean ± s.e.m. Data sets are compared by *t*-test. No significant differences were detected

anaesthetised mice[15] incrementally depolarized the membrane potential in a Piezo1-dependent manner (Fig. 5c).

The observations suggested a type of Piezo1 channel which is sufficiently abundant and active to strongly regulate the resting membrane potential. It is a specialised non-inactivating form of Piezo1 channel but otherwise it has the expected unitary conductance and pharmacology. The channel is strongly constitutively active but can further activate in proportion to fluid flow. The commonly reported inactivation of exogenously over-expressed Piezo1 channels has previously been found to be lost after pulses of mechanical force and suppressed by application of force to extracellular domains without effect on channel unitary conductance[16, 17]. There is therefore precedence for Piezo1 channels adopting a non-inactivating state.

**Coupling to voltage-gated Ca$^{2+}$ entry in smooth muscle cells**. To understand how the Piezo1 mechanism could become relevant at times of whole body physical activity, we hypothesized that the increased blood flow in exercise drives depolarization which has been shown previously to be efficiently coupled to the adjacent vascular smooth muscle cells[10, 18, 19] and could be sufficient to activate pro-contractile voltage-gated Ca$^{2+}$ channels[20] only when the depolarization reaches a particular range of values. To test this hypothesis, we recorded from vascular smooth muscle cells freshly isolated from second-order mesenteric artery. Increasing depolarizations were applied by voltage-clamp to activate voltage-gated Ca$^{2+}$ currents. As expected these currents were small, close to the technical limits of detection (Fig. 5d). Such channels have no distinct threshold for activation but show exponential

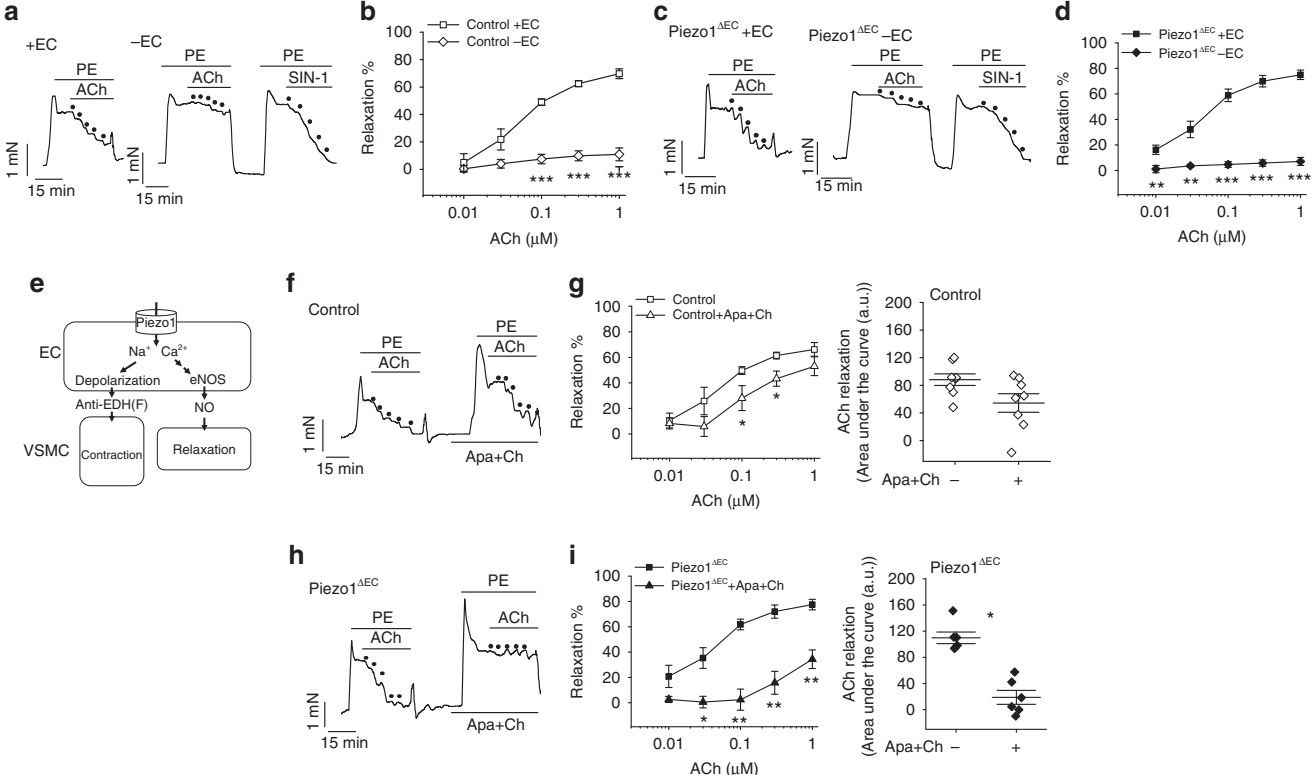

**Fig. 2** Endothelial Piezo1 channels have an anti-EDH(F) effect. Isometric tension recordings from mouse second-order mesenteric artery. **a** Example recordings from control genotype artery before (+ EC) and after endothelium-denudation (−EC). Upward deflection is increasing tension. Phenylephrine (PE, 0.3 μM). Acetylcholine (ACh) and the nitric oxide donor amino-3-morpholinyl-1,2,3-oxadiazolium (SIN-1) were applied at increasing concentrations as indicated by the dots (0.01, 0.03, 0.1, 0.3 and 1 μM). **b** As for **a** but mean data (*n* = 14 mice). **c**, **d** As for **a**, **b** but Piezo1$^{\Delta EC}$ mice (*n* = 10 mice). **e** Schematic illustration of the dichotomy Piezo1 presents for endothelial biology and vascular function. EC, endothelial cell. VSMC, vascular smooth muscle cell. eNOS, endothelial nitric oxide synthase. NO, nitric oxide. EDH(F), endothelium-derived hyperpolarization (factor). **f** Example recordings from control genotype artery before and after application of apamin (Apa, 0.5 μM) and charybdotoxin (Ch, 0.1 μM). Phenylephrine (PE, 0.3 μM). Acetylcholine (ACh) was applied at increasing concentrations as indicated by the *dots* (0.01, 0.03, 0.1, 0.3 and 1 μM). **g** As for **f** but mean data (*n* = 8 mice). **h**, **i** As for **f**, **g** but for Piezo1$^{\Delta EC}$ mice (*n* = 6 mice). Averaged data are displayed as mean ± s.e.m. Data sets are compared by *t*-test. Statistical significance is indicated by *$P < 0.05$, **$P < 0.01$, ***$P < 0.001$

increases in opening probability with depolarization[20]. To take account of this behaviour, we fitted the data with the Boltzmann equation, leading to the suggestion that increases in Ca²⁺ entry occurred progressively from −40 mV to more positive (Fig. 5e). In the whole body, the cell-rich fluid of blood causes shear stress at the endothelium and whole body physical activity increases shear stress[21]. We demonstrated the principle that increasing fluid flow, and thus shear stress, caused depolarization which reached the activation range for voltage-gated Ca²⁺ channels (Fig. 5e). To test this idea in the intact artery, we cannulated segments of second-order mesenteric artery to apply pressure and flow to the lumen. Importantly, increased flow caused vasoconstriction which was inhibited by nicardipine, a blocker of voltage-gated Ca²⁺ channels (Fig. 6a–c). In Piezo1$^{\Delta EC}$ mice, flow-induced vasoconstriction was absent (Figs. 6d, e). These data support a hypothesis whereby precipitous increases in voltage-gated Ca²⁺ entry occur in smooth muscle cells, leading to vasoconstriction when fluid flow along the endothelial surface is sufficiently high.

There is absence of specific information on the absolute shear stress experienced by endothelial cells in second-order mouse mesenteric arteries at rest or during whole body physical activity. In vivo, shear stress is complicated by the pulsating cardiac rhythm, vascular architecture and arterial calibre, viscosity and cellular content of the blood, and glycoprotein-polysaccharide structures between blood and membrane proteins of the endothelial cells. Nevertheless, shear stress in mice is considered

to range from about 3 to 60 Pa[22–24], consistent with our studies (Fig. 5e). Values in humans are usually lower[22, 23].

**Endothelial Piezo1 is important for physical performance.** Based on the above findings, we hypothesised a Piezo1 mechanism which senses whole body physical activity in order to constrict mesenteric resistance arteries with the purpose of directing mesenteric blood flow away from the gastrointestinal tract[25, 26] to other organs—in particular skeletal muscle—to improve physical performance. To test this idea, we quantitatively investigated running wheel performance. Although the Piezo1$^{\Delta EC}$ mice were superficially normal (Fig. 1), their performance was compromised (Fig. 7a–d) (Supplementary Fig. 8). The impact of Piezo1$^{\Delta EC}$ declined with continued exposure to the wheel, suggesting compensated performance due to physical training (Fig. 7a–c) despite the blood pressure lowering effect being sustained (Fig. 3b). Although Piezo1$^{\Delta EC}$ mice gained weight normally in the absence of the wheel (Figs 1b and 7e), they lost more weight than controls once they had access to the wheel, suggesting that they were working harder to achieve their expectations (Fig. 7e).

Therefore, we suggest a molecular mechanism for sensing whole body physical activity: specialised Piezo1 channels in the endothelium which beneficially impact on overall physical performance. We outline a mechanistic principle by which the

channels can sense fluid flow and transduce it into constriction of mesenteric arteries and thus increase blood pressure during physical activity because these arteries contribute a major component of total peripheral resistance.

**Contribution of endothelial Piezo1 is vascular bed specific.** It is important to recognise that different vascular beds respond differently to whole body physical activity. While blood flow to the intestines decreases in physical activity[25, 26], blood flow to skeletal

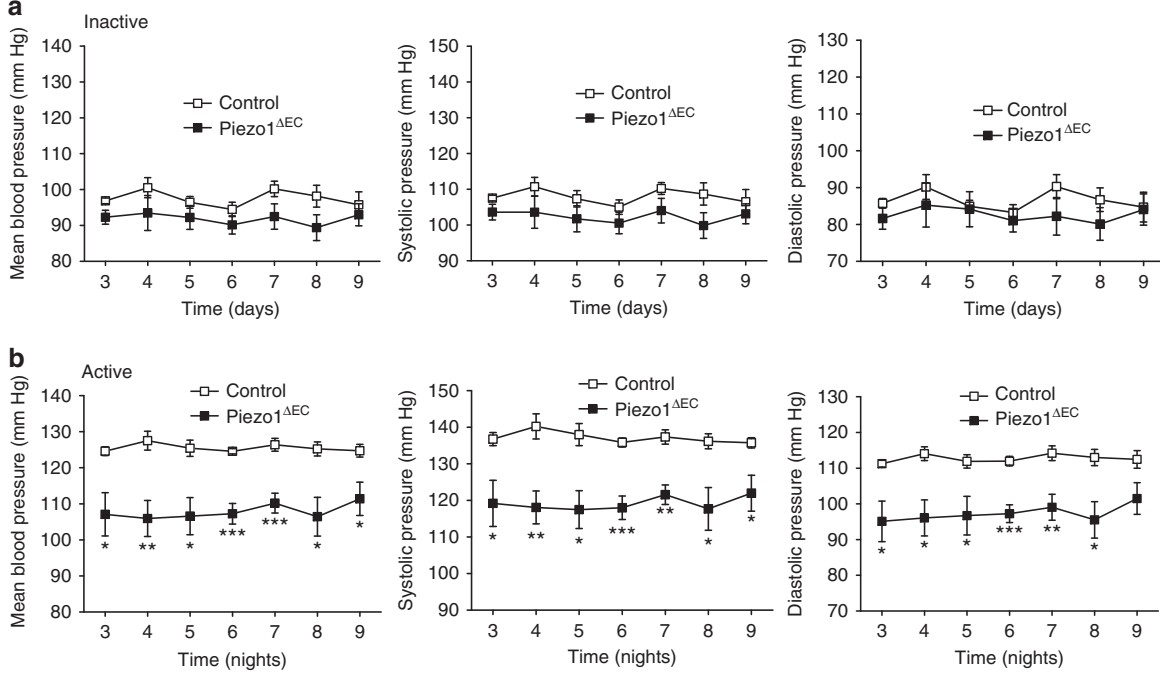

**Fig. 3** Importance for elevated blood pressure during whole body physical activity. Telemetry measurements of mean, systolic and diastolic blood pressures in conscious freely moving control ($n = 6$) and Piezo1$^{\Delta EC}$ ($n = 7$) mice. Data were analysed when the mice were inactive during the day **a** and voluntarily active on a running wheel during the night **b**. Time zero is when the mouse was introduced to the running wheel cage. Measurements were not made during the first 2 days of acclimatization. Averaged data are displayed as mean ± s.e.m. Data sets are compared by $t$-test. Statistical significance is indicated by *$P < 0.05$, **$P < 0.01$, ***$P < 0.001$

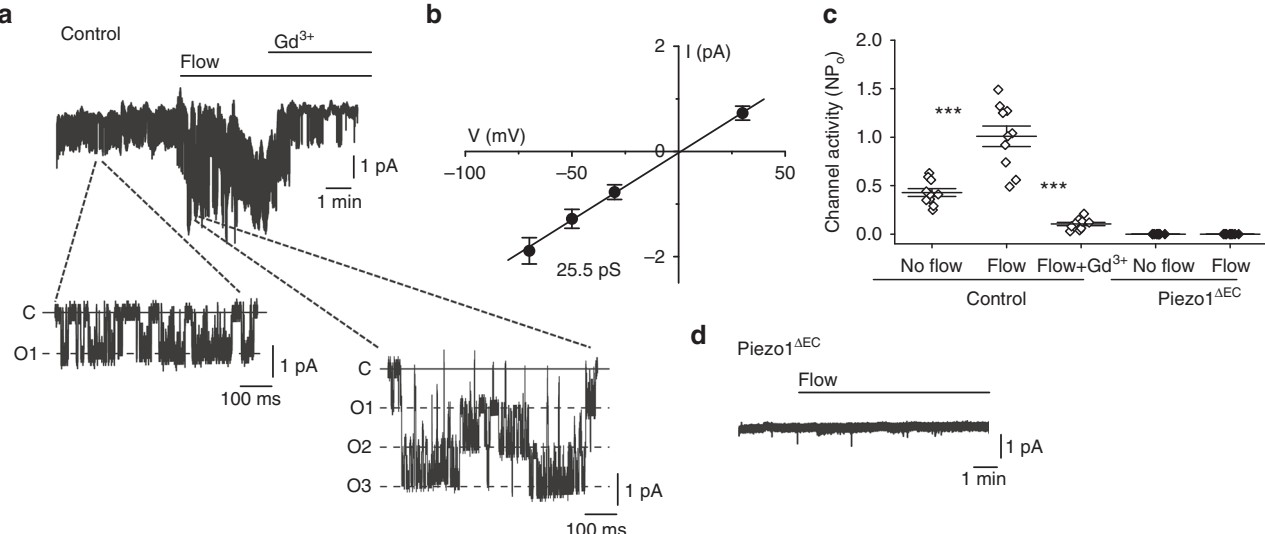

**Fig. 4** Piezo1 channels are flow sensors in endothelium of mesenteric resistance artery. Data are for ionic current recordings from outside-out patches excised from freshly isolated endothelium of second-order mesenteric arteries. **a** Example recording at −70 mV. Two sections are expanded to clarify unitary current events (C: channel closed) (O: 1, 2 or 3 simultaneous channel openings). The patch was placed at the outlet of a capillary from which flowed ionic solution at 20 μl s$^{-1}$. Gadolinium ion (Gd$^{3+}$, 10 μM). **b** Mean unitary current amplitudes for channels activated by flow as in **a** ($n = 10$). **c** Mean channel activity (NP$_o$: number × probability of opening) for experiments of the type exemplified in **a** for no flow and flow conditions and the two genotypes (Control and Piezo1$^{\Delta EC}$). Individual data points for each independent experiment are shown as *symbols*, superimposed on which are the mean ± s.e.m. values ($n = 10$ for each group). **d** Example original trace for a patch from Piezo1$^{\Delta EC}$ endothelium exposed to 20 μl s$^{-1}$ flow. Averaged data are displayed as mean ± s.e.m. Data sets are compared by $t$-test. Statistical significance is indicated by ***$P < 0.001$

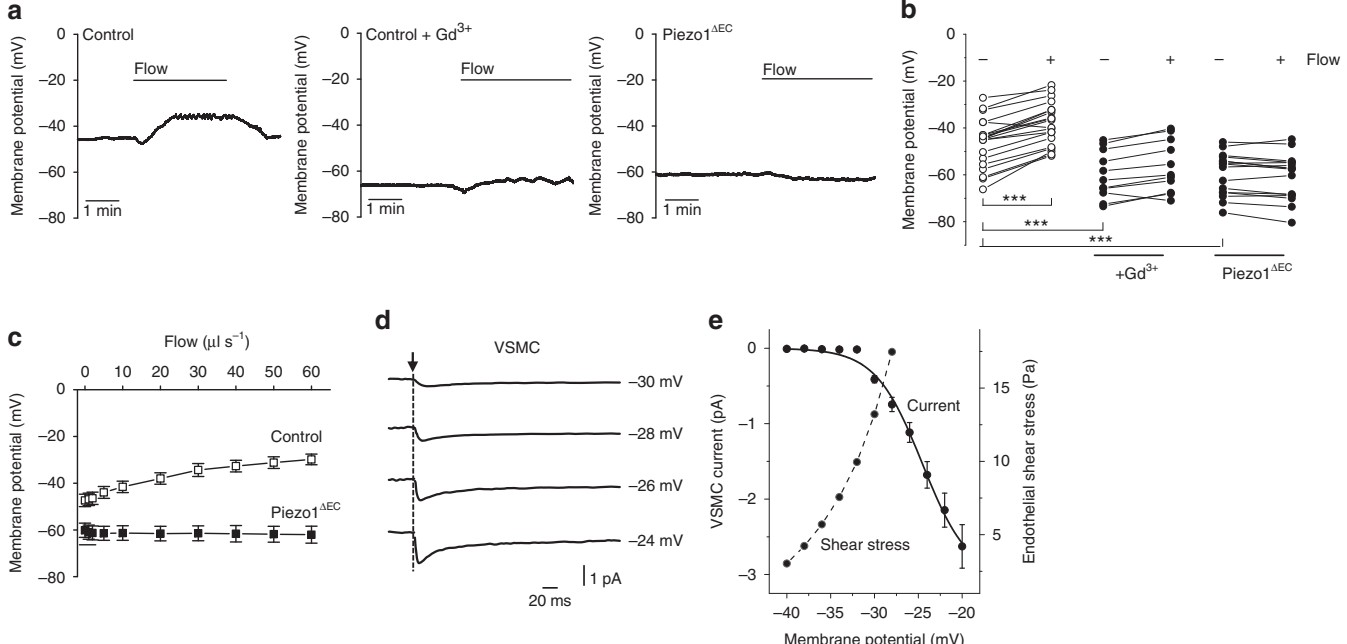

**Fig. 5** $Ca^{2+}$ channel activation in vascular smooth muscle cells. **a, b** Membrane potential measurements from freshly isolated endothelium of second-order mesenteric arteries. **a** Example traces from the control and Piezo1$^{\Delta EC}$ genotypes in the absence of $Gd^{3+}$ and the control genotype in the presence of 10 μM $Gd^{3+}$. Endothelium was exposed to flow at 20 μl s$^{-1}$. **b** As for **a** but individual data points for the three independent experiments shown as symbols. Control genotype: no flow $-46.3 \pm 2.4$ mV *vs* flow $-38.4 \pm 2.1$ mV (***). Control genotype in $Gd^{3+}$: no flow $-60.1 \pm 3.1$ mV *vs* flow $-56.5 \pm 3.3$ mV (***). Piezo1$^{\Delta EC}$ genotype: no flow $-59.5 \pm 2.3$ mV *vs* flow $-60.7 \pm 2.5$ mV (*). **c** As for **a, b** but a separate series of experiments in which endothelium was exposed to increasing flow in the absence of $Gd^{3+}$ for control ($n = 10$) and Piezo1$^{\Delta EC}$ ($n = 6$) genotypes. **d** Example current recordings from vascular smooth muscle cells (VSMC) freshly isolated from second-order mesenteric artery. Square-step depolarizations were applied at the time of the arrow from a holding voltage of $-80$ mV to the test voltage indicated. Linear leak and capacitance currents were subtracted. **e** Mean data for peak VSMC inward currents of the type exemplified in **d** (current, $n = 8$). Superimposed in *grey* are endothelial shear stress values calculated from the control genotype data in **c**. Membrane potential for the highest shear stress was obtained by extrapolation based on a least-squares fit of the Hill equation. Averaged data are displayed as mean $\pm$ s.e.m. Data sets are compared by *t*-test. Statistical significance is indicated by ***$P < 0.001$

muscle increases[2], so a vasodilator rather than vasoconstrictor mechanism must dominate in resistance arteries of skeletal muscle[2]. Similarly, blood flow to the brain is maintained or slightly increased during physical activity to avoid syncope[2]. Therefore, we investigated the significance of endothelial Piezo1 in saphenous artery, which supplies the leg, and carotid artery, which supplies the brain. Saphenous artery was similar in calibre to second-order mesenteric artery: its external diameter when pressurized to 60 mm Hg was $284.2 \pm 13.4$ μm ($n = 15$) compared with $279.8 \pm 12.7$ μm in mesenteric artery ($n = 17$). Carotid artery was $428.8 \pm 17.4$ μm under similar conditions ($n = 8$). In contrast to mesenteric artery (Fig. 2) the EDH(F) effect was not amplified by Piezo1 disruption in saphenous or carotid artery (Fig. 8a–h) (Supplementary Fig. 9). The data suggest vascular bed specificity of Piezo1 in a way which enables it to enhance physical performance.

## Discussion

We have described the challenge presented by the $Ca^{2+}$ and depolarizing effects of Piezo1 channels in endothelial cells and suggested a role for the depolarising effect in whole body physical activity. Here, however, we have not addressed the relationship between Piezo1 and nitric oxide synthase and nitric oxide suggested previously[7, 13]. When we studied flow-induced vasoconstriction, an inhibitor of endothelial nitric oxide synthase was present in order to focus on the Piezo1 depolarization mechanism (Fig. 6). Importantly, the Piezo1-dependence of increased blood pressure in whole body physical activity suggests dominance of

the vasoconstrictor mechanism in vivo even without exogenous nitric oxide synthase inhibition (Fig. 3b). This implies that the nitric oxide mechanism is naturally suppressed in mesenteric arteries and perhaps also in other arteries which constrict during whole body physical activity such as those of the kidneys and liver[2]. A flow-induced vasodilatation mechanism exists in mesenteric arteries[13, 27] but our data suggest that it is less important than vasoconstriction during whole body physical activity. The vasodilatory effect might of course be important in other circumstances. The role of endothelial Piezo1 has been studied in sedate mice where relevance of the nitric oxide mechanism has been suggested[13]. The effect reported was however on systolic blood pressure, with no diastolic data shown[13]. Because vasodilatation should affect diastolic pressure, it remains to be clarified if endothelial Piezo1 is important for vasodilation in vivo in sedentary mice. Our data suggest no role of endothelial Piezo1 when mice are inactive between periods running on a wheel (Fig. 3a).

The principles we describe here in the mouse may be important in people because Piezo1 is a functional endothelial protein in humans[28] and whole body physical activity increases vascular wall shear stress in humans[21]. In healthy individuals, systolic blood pressure usually increases due to increased cardiac output whereas diastolic blood pressure may remain unchanged depending on the extent of vasodilation in skeletal muscle[2]. Nevertheless, whole body physical activity redistributes blood away from the intestines, suggesting importance of vasoconstriction in the mesenteric bed[25, 26]. Exercise is commonly used as an approach for protecting against or reducing hypertension in

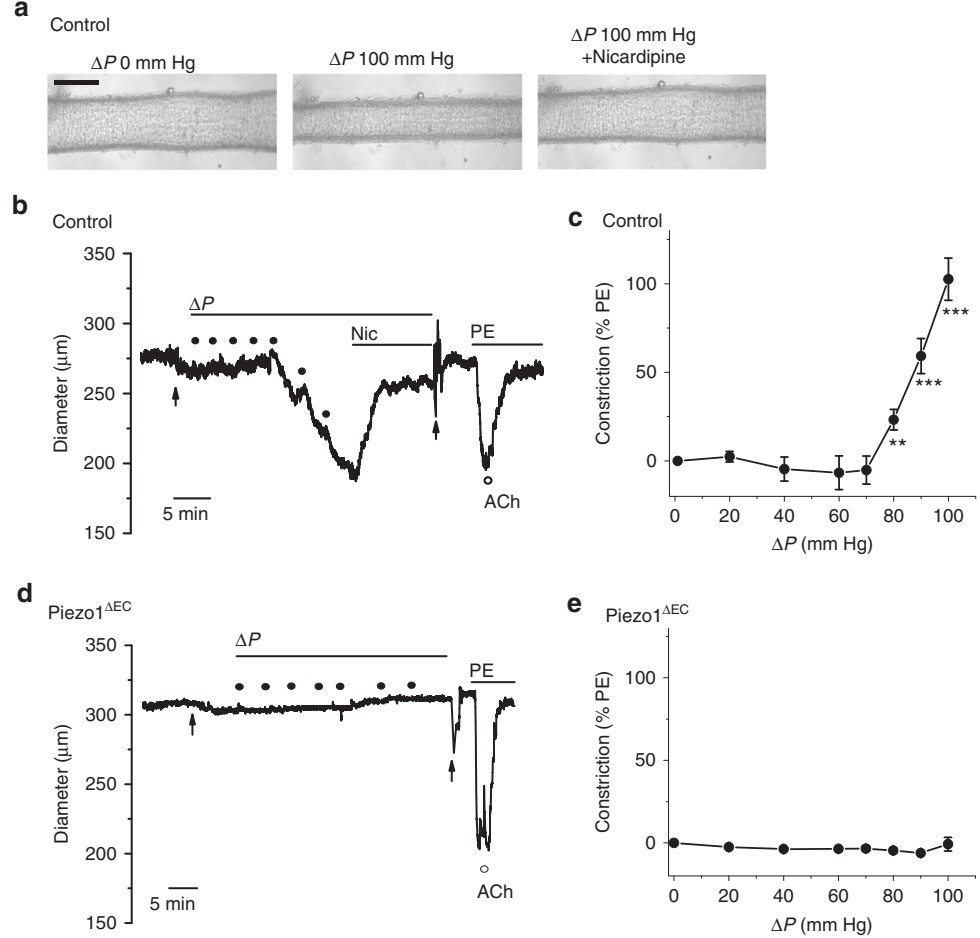

**Fig. 6** Flow-evoked vasoconstriction in mesenteric artery. Isobaric external diameter recordings from second-order mesenteric artery. **a** Example images of a cannulated artery before and after luminal pressure difference ($\Delta P$) and then after 10 μM nicardipine was added to the recording chamber. Control genotype mouse. *Scale bar*, 200 μm. **b** Example diameter recording for a control genotype mouse during incremental increases in $\Delta P$ as indicated by the *black dots* (20, 40, 60, 70, 80, 90 and 100 mm Hg). Nicardipine (Nic, 10 μM), phenylephrine (PE, 1 μM) and acetylcholine (ACh, 10 μM) were applied as indicated. The first *arrow* indicates addition of 100 μM N(ω)-nitro-L-arginine methyl ester (L-NAME) to the recording chamber and the second arrow multiple washes of the chamber to remove nicardipine and L-NAME. **c** Mean data for the type of experiment shown in **b** presented as the constriction to $\Delta P$ as a percentage of the PE response (9 arteries from $n = 6$ mice). Nicardipine significantly (***) reduced the 100 mm Hg $\Delta P$ response to $30.7 \pm 3.9\%$ ($n = 6$ mice). **d**, **e** The same as for **b**, **c** but using Piezo1$^{\Delta EC}$ genotype mice (8 arteries from $n = 3$ mice). Averaged data are displayed as mean ± s.e.m. Data sets are compared by t-test. Statistical significance is indicated by **$P < 0.01$, ***$P < 0.001$

the human population[29] and, conversely, exercise-induced hypertension is suggested as an important predecessor of persistent hypertension, which remains one of the major health concerns of the twenty-first century[30]. Therefore developing further methodologies and tools for studying Piezo1 and its relationships to exercise-induced adaptations should be a valuable area for future research.

## Methods

**Piezo1-modified mice**. All animal use was authorized by the University of Leeds Animal Ethics Committee and The Home Office, UK. All animals were maintained in GM500 individually ventilated cages (Animal Care Systems), except during telemetry recordings, at 21 °C 50–70% humidity, light/dark cycle 12/12 h on RM1 diet (SpecialDiet Services, Witham, UK) ad libitum and bedding of Pure'o Cell (Datesand, Manchester, UK). Genotypes were determined using real-time PCR with specific probes designed for each gene (Transnetyx, Cordova, TN). C57BL/6 J mice with *Piezo1* gene flanked with LoxP sites (Piezo1$^{flox}$) were described previously[7]. To generate tamoxifen (TAM) inducible disruption of *Piezo1* gene in the endothelium, Piezo1$^{flox}$ mice were crossed with mice expressing cre recombinase under the Cadherin5 promoter (Tg(Cdh5-cre/ERT2)1Rha) and inbred to obtain Piezo1$^{flox/flox}$/Cdh5-cre mice. TAM (Sigma-Aldrich) was dissolved in corn oil (Sigma-Aldrich) at 20 mg ml$^{-1}$. Mice were injected intra-peritoneal with 75 mg kg$^{-1}$ TAM for 5 consecutive days and studies were performed 10–14 days later. Piezo1$^{flox/flox}$/Cdh5-cre mice that received TAM injections are referred to as

Piezo1$^{\Delta EC}$. Piezo1$^{flox/flox}$ littermates (lacking Cdh5-cre) that received TAM injections were the controls (control genotype). For experiments, mice were males aged 12–16 weeks, except for telemetry (14–18-week old) and femoral artery injury (18–22-week old).

**Analysis of Piezo1 deletion**. Samples (about 10 mm$^3$) of liver, lung, aorta, femoral artery and mesenteric artery were digested overnight at 37 °C in a lysis buffer containing 10 mM Tris pH 7.4, 50 mM EDTA, 1 % SDS; 5 μg ml$^{-1}$ proteinase K (Sigma-Aldrich). Samples were then vortexed and 400 μl of phenol/chloroform/isoamyl (Sigma-Aldrich) was added. Tubes were mixed by inverting/shaking 20 times every 15 min for 1 h then centrifuged at 13,000×g for 15 min at room temperature. Four-hundred microliter of the top layer was transferred to a new tube followed by an addition of 440 μl of isopropanol and 40 μl of 3 M NaCl. Tubes were mixed by gentle inverting and left to stand for 1 h at room temperature. DNA was pelleted by centrifugation at 13,000×g for 30 min at room temperature. The supernatant was discarded and the pellet was washed with 70% ethanol, briefly air dried and resuspended in 60 μl of TE buffer. DNA was amplified using 12.5 μl Bioline MyTaq Red Mix, 0.5 μl of DNA solution, 1 μM of each primer. Sequences of PCR primers are specified in Supplementary Table 1. PCR was 95 °C for 5 min; 32 cycles of 95 °C for 30 s, 60 °C for 30 s, 72 °C for 30 s; 72 °C for 5 min. PCR products were electrophoresed on 2 % agarose gels containing SYBR safe (Roche) at 80 V for 45 min.

**Mouse liver endothelial cells**. Mouse liver sinusoidal endothelial cells were isolated using an immunomagnetic separation technique. A whole mouse liver was

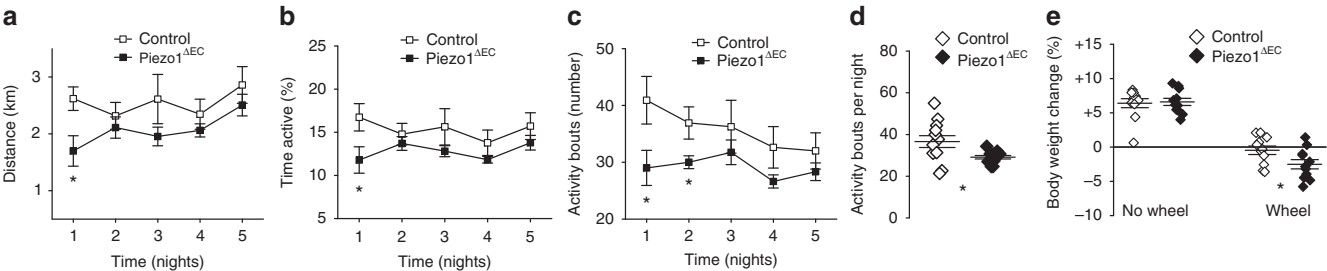

**Fig. 7** Physical performance depends on endothelial Piezo1. Voluntary running wheel data for control and Piezo1$^{\Delta EC}$ genotypes from the dark cycle (active period) showing distance run on the wheel **a**, percentage time for which mice were active on the wheel **b** and the number of active bouts of exercise **c** (*n* = 12 mice per group). **d** As for **c** but summary analysis for all bouts of activity. **e** For the same mice analysed in **a**–**d**, changes in body weight during the 12 days prior to and 7 days after access to the wheel (*n* = 12 mice per group). Averaged data are displayed as mean ± s.e.m. Data sets are compared by *t*-test. Statistical significance is indicated by **P* < 0.05

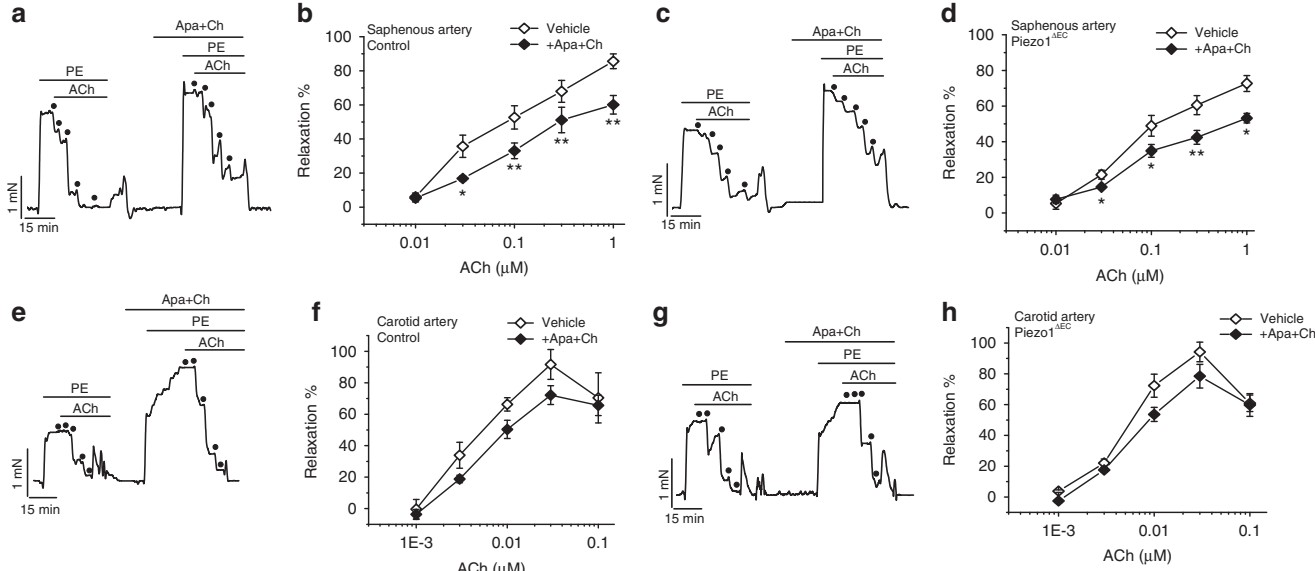

**Fig. 8** Contribution of endothelial Piezo1 is vascular bed specific. Isometric tension recordings from mouse saphenous or carotid artery. **a** Example recordings from control genotype saphenous artery before and after application of apamin (Apa, 0.5 μM) and charybdotoxin (Ch, 0.1 μM). Phenylephrine (PE, 0.3 μM). Acetylcholine (ACh) was applied at increasing concentrations as indicated by the *dots* (0.01, 0.03, 0.1, 0.3 and 1 μM). **b** As for **a** but mean data (*n* = 5 mice). **c**, **d** As for **a**, **b** but for Piezo1$^{\Delta EC}$ mice (*n* = 5 mice). **e**–**h** As for **a**–**d** but carotid artery (*n* = 5 control mice, *n* = 5 Piezo1$^{\Delta EC}$ mice). Averaged data are displayed as mean ± s.e.m. Data sets are compared by *t*-test. Statistical significance is indicated by **P* < 0.05, ***P* < 0.01

minced using 2 scalpel blades and resuspended in a dissociation solution consisting of 9 ml 0.1 % collagenase II, 1 ml 2.5 U ml$^{-1}$ dispase, 1 μM CaCl$_2$ and 1 μM MgCl$_2$ in Hanks Buffer solution. The tissue-dissociation mix was incubated at 37 °C for 50 min in a MACSMix Tube Rotator (Miltenyi Biotech) to provide continuous agitation. At the end of enzymatic digestion the sample was passed through 100 and 40 μm cell strainers to remove any undigested tissue. Cells were washed twice in PEB buffer consisting of Phosphate Buffered Saline (PBS), EDTA 2 mM and 0.5% Bovine Serum Albumin (BSA), pH 7.2. The washed pellets were resuspended in 1 ml of PEB buffer and 200 μl of dead cell removal paramagnetic microbeads per 1 × 10$^7$ cells (Miltenyi Biotec) at room temperature for 15 min. After incubation the cells were passed through an LS column prepared with 1 × binding buffer (Miltenyi Biotec) in a magnetic field (MiniMACS Separator, Miltenyi Biotec). The eluate was then incubated with 20 ml red blood cell lysis buffer consisting of 0.206 g Tris base, 0.749 g NH$_4$Cl in 100 ml PBS pH to 7.2. Cells were washed again in PEB buffer, and the pellet was resuspended in 1 ml PEB buffer and 30 μl CD146 microbeads (Miltenyi Biotec) at 4 °C for 15 min under continuous agitation. After incubation this solution was passed through an MS column prepared with PEB buffer. CD146 positive cells were retained in the column and CD146 negative cells passed through as eluate. CD146 positive cells were washed through with warm EGM-2 media and the CD146 selection process was repeated a second time. After a second purification cells were plated and grown in a 5% CO$_2$ incubator at 37 °C. Media were changed at 12 h and then every 24 h until confluent.

**Piezo1 inducible cell line**. Piezo1-GFP[7] was used as a PCR template to clone human Piezo1 coding sequence into pcDNA™4/TO between HindIII and EcoRI

restriction sites. Piezo1 was amplified as two fragments using the following primers: (HindIII-Piezo1-Fw: AATAAGCTTATGGAGCCGCACGTG and BamHI-Int.Piezo1-Rv: AATGGATCCCCCTGGACTGTCG) and (BamHI-Int.Piezo1-Fw: AATGGATCCTCCCCGCCACGGA and EcoRI-Piezo1-Rv: AATGAATTCTTACTCCTTCTCACGAGT). The two fragments were fused using BamHI restriction site, resulting in the full length Piezo1 coding sequence with the c4182a silent mutation. T-RExTM-293 cells were transfected with pcDNA4/TO-Piezo1 using Lipofectamine 2000 (Thermo Fisher Scientific). Subsequently cells were treated with 10 μg ml$^{-1}$ blasticidin and 200 μg ml$^{-1}$ zeocin (Invitrogen, Thermo Fisher Scientific) to select stably transfected cells. Single cell clones were isolated and analysed individually. Expression was induced by treating the cells for 24 h with 10 ng ml$^{-1}$ tetracycline (Sigma-Aldrich) and analysed by quantitative RT-PCR and western blot.

**Fura-2 Ca$^{2+}$ measurements**. Intracellular Ca$^{2+}$ was measured using the ratiometric Ca$^{2+}$ indicator dye fura-2. Experiments were performed on confluent cells in a 96-well plate. Cells in each well were incubated with 50 μl fura-2 AM loading solution for 1 h at 37 °C. The loading solution consisted of 2 μM fura-2 AM and 0.01% pluronic acid in Standard Bath Solution (Ca$^{2+}$-SBS) consisting of 130 mM NaCl, 5 mM KCl, 1.2 mM MgCl$_2$, 1.5 mM CaCl$_2$, 8 mM D-glucose and 10 mM HEPES (pH 7.4). After 1 h the loading solution was removed and 100 μl of Ca$^{2+}$-SBS was added to each well and left at room temperature for 30 min. A compound plate was prepared at twice the final concentration tested in Ca$^{2+}$-SBS. The FlexStation II$^{384}$ was set to add 80 μl of the compound solution to each well on the test plate containing 80 μl of Ca$^{2+}$-SBS. Baseline fluorescence ratios were

recorded before addition of the compound solution to the cell plate after 60 s, with regular recordings thereafter for a total of 5 min.

**Thallium FluxOR measurements**. Cells were plated at 80–90% confluence in 96-well plates 24 h prior to recordings ($5 \times 10^4$ Human Embryonic Kidney (HEK) T-REx cells; $1.92 \times 10^4$ Human Umbilical Vein Endothelial Cells (HUVECs)). HEK T-Rex cells were from Thermo Fisher Scientific (catalogue #R71007) and HUVECs were from Lonza (catalogue #CC-2519). HEK T-Rex cells were validated to be tetracycline-responsive (as expected) (Supplementary Fig. 6). HUVECs were validated by positive staining with anti-CD31 antibody, response to vascular endothelial growth factor, and alignment of the cells to shear stress; human nucleotide sequences were detected, confirming human origin. To measure thallium ($Tl^+$) influx, cells were loaded with the FluxOR$^{TM}$ dye for 1 h at room temperature, transferred to assay buffer and stimulated with a $Tl^+$ containing $K^+$-free solution as per the manufacturer's instructions (Molecular Probes). Measurements were made on a fluorescence plate reader (FlexStation II$^{384}$). FluxOR was excited at 485 nm and emitted light collected at 520 nm, measurements expressed as a ratio increase over baseline ($F/F_0$), with vehicle (DMSO) values subtracted from Yoda1 values at each time point ($\Delta F/F_0$). Rates of increase in fluorescence intensity were determined between 7.5 and 35 s after injection of Yoda1 ($\Delta F/F_0/ms$).

**Arterial contraction studies**. Animals were culled by cervical dislocation according to Schedule 1 procedure approved by the UK Home Office. Mesenteric arcades were dissected out and placed immediately into ice-cold Krebs solution (125 mM NaCl, 3.8 mM KCl, 1.2 mM $CaCl_2$, 25 mM $NaHCO_3$, 1.2 mM $KH_2PO_4$, 1.5 mM $MgSO_4$, 0.02 mM EDTA and 8 mM D-glucose, pH 7.4). Second-order mesenteric, saphenous or carotid arteries were cleaned of fat and connective tissue under a dissection microscope. Segments of 1 mm length were mounted in an isometric wire myograph (Multi Wire Myograph System, 620 M), bathed with Krebs solution warmed at 37 °C and gassed with 95% $O_2$/5% $CO_2$ then stretched stepwise radially to their optimum resting level to an equivalent trans-mural pressure of 100 mm Hg and equilibrated for 1 h prior to experiments. For studies of luminal flow in second-order mesenteric artery, vessel segments were mounted on glass cannulas in a pressure myograph (Model 110p, Danish Myo Technology A/S, Denmark). Flow was generated by increasing the pressure difference ($\Delta P$) between inflow and outflow without change in the absolute intraluminal pressure. The outer arterial diameter was monitored using a CCD camera (DMX41 AU02, Imaging Source Europe, Germany) and recorded with MyoView II software. Arteries were only used for investigation if they constricted in response to phenylephrine (PE) and dilated in response to acetylcholine (ACh).

**Blood pressure measurements**. Conscious long-term recordings of arterial blood pressure (mean, systolic and diastolic) were achieved via a radiotelemetry probe (model TA11PA-C10, Data Sciences International). Adult male mice (14–18-week old) were anaesthetised with isoflurane (5% induction 1.5% maintenance) in 95% $O_2$ and body temperature maintained via a heating pad. The probe catheter was advanced, via the left carotid artery, into the ascending aorta. The body of the transmitter was placed in a subcutaneous pocket along the left flank. A period of at least 14 days was allowed for recovery from surgery before the start of experimental recordings. TAM treatment started 4 days after probe implantation and recordings started 10 days after the last TAM injection. Mice were housed singly in cages and synchronized to a light–dark cycle of 12:12 h with lights on at 06:00 h. Cages were positioned over receivers connected to a computer system for data recording. Blood pressure waveforms and parameters were analysed using DSI analysis package, Dataquest ART 4.1. Continuous 24 h recordings were begun 3 days following singular housing and obtained over a 7 day period. During the recording period animals were allowed free access to a voluntary running wheel.

**Freshly isolated mesenteric endothelial cells**. Endothelial cells were freshly isolated from second-order branches of mouse mesenteric arteries as described previously[14]. Briefly, dissected second-order mesenteric arteries were enzymatically digested in dissociation solution (126 mM NaCl, 6 mM KCl, 10 mM Glucose, 11 mM HEPES, 1.2 mM $MgCl_2$, 0.05 mM $CaCl_2$, with pH adjusted to 7.2) containing 1 mg ml$^{-1}$ collagenase Type IA (Sigma-Aldrich, Dorset, UK) for 14 min at 37 °C and then triturated gently to release the endothelial cells on a glass coverslip.

**Patch-clamp electrophysiology**. Membrane potential was measured using the perforated whole-cell configuration of the patch-clamp technique in current clamp mode with an Axopatch-200A amplifier (Axon Instruments, Inc.) equipped with Digidata 1440 A and pCLAMP 10.6 software (Molecular Devices, Sunnyvale, CA, USA) at room temperature. Outside-out membrane patch recordings were made using the same equipment but in voltage-clamp mode. Endothelial cells and endothelium were bathed in a solution consisting of 135 mM NaCl, 4 mM KCl, 2 mM $CaCl_2$, 1 mM $MgCl_2$, 10 mM glucose and 10 mM HEPES (pH 7.4). Heat-polished patch pipettes with tip resistances between 3 and 5 MΩ were used. For

membrane potential recordings, amphotericin B (Sigma-Aldrich) was used as the perforating agent, added in the pipette solution composed of 145 mM KCl, 1 mM $MgCl_2$, 0.5 mM EGTA and 10 mM HEPES (pH 7.2). For application of fluid flow, endothelium or membrane patches were manoeuvred to the exit of a capillary tube with tip diameter of 350 μm, out of which ionic (bath) solution flowed at rates specified in the main text and figure legends. Calculation of shear stress ($\tau_\omega$) was achieved using the Hagen-Poiseuille formula[31] ($\tau_\omega = 4\mu Q/\pi R^3$) where $\mu$ is dynamic viscosity, $Q$ is flow rate and $R$ is radius of the capillary tube.

**Echocardiography**. Animals were maintained under steady-state isofluorane anaesthesia and placed on a heated platform with ECG and respiration monitoring. Core temperature was measured using a rectal probe (Indus Instruments) and maintained at 37.5 °C throughout recording. Echocardiography was performed using a Vevo2100 high resolution, pre-clinical in vivo ultrasound system (Visual-Sonics) with the MS-550D transducer at 40 MHz frequency and 100% power. Imaging was performed on a layer of aquasonic gel after the pre-cordial skin had been clipped and de-epliated with cream (Veet). Parasternal long-axis view (PLAX) images were obtained in EKV mode (set at 1000 Hz for recording) over the entire cardiac cycle. The left ventricular area was traced in end-distole (LVAd) and end-systole (LVAs) and used to derive the ejection fraction (EF) with the Vevo LAB cardiac package software. The investigator performing sonography was blinded to the genotype of the animals. Transverse EKV recordings were also obtained over the abdominal aorta just below the diaphragm using the same settings as described for the heart. These images were evaluated in the VevoVasc software package to determine vessel distensibility. Maximal anteroposterior aortic diameter (from inner wall to inner wall) was measured in the same images in systole and diastole using Vevo LAB general imaging package software.

**Retina vasculature staining and analysis**. Retinas were dissected from eyes after fixation in 4% paraformaldehyde in PBS for 4 h at room temperature, then stored overnight at 4 °C in permeabilisation and blocking buffer (PBS; 0.5% triton; 1% BSA; 0.01% sodium deoxycholate; 0.02% sodium azide; 0.1 mM $CaCl_2$; 0.1 mM $MgCl_2$; 0.1 mM $MnCl_2$). Retinal vasculature was then stained overnight at 4 °C with isolectin B4 Alexa Fluor 488 conjugate (Molecular Probes, Thermo Fisher), diluted 1:100 in PBLEC buffer (PBS; 1% triton; 0.1 mM $CaCl_2$; 0.1 mM $MgCl_2$; 0.1 mM $MnCl_2$). Retinas were washed with 0.25% Triton in PBS, then flat-mounted on slides with ProLong Gold (Molecular Probes, Thermo Fisher). Confocal microscopy (LSM 880, Zeiss) was used to image retinas, with analysis blinded to genotype conducted using ImageJ software (NIH, Bethesda, MD). Distal arterial diameter was analysed 1500 μm from the optic disc in the largest branch of each artery emanating from the disc. Capillary area was determined in regions of interest, which excluded arteries and veins, using the threshold function and fractional area measurement.

**Femoral injury**. Experiments were carried out on 18–22-week-old male mice. Femoral injury was performed 12–16 days after the last TAM injection. Mice were anesthetized with isoflurane (1.5–2%) before a small incision was made in the mid-thigh and extended. Having carefully isolated the femoral artery, an arteriotomy was made in the saphenous artery using iris scissors (World-Precision Instruments, Sarasota, FL) and a 0.014-inch-diameter angioplasty guidewire with tapered tip (Hi-Torque Cross-It 200XT, Abbott-Vascular, IL) was introduced. The guidewire was advanced 1.5 cm in to the femoral artery, and three passages performed per mouse, resulting in complete endothelial denudation. The guidewire was removed completely and a suture tightened rapidly immediately distal to the bifurcation of the femoral artery. The skin was closed with a continuous suture. Animals received peri-operative analgesia with buprenorphine (0.25 mg kg$^{-1}$ s.c.). Mice were anes-thetized at 5 days after wire injury and 50 μl of 0.5% Evans blue dye injected into the inferior vena cava. The mice were perfused/fixed with 4% paraformaldehyde in PBS before the femoral arteries were collected. The vessels were opened long-itudinally. The areas stained and unstained in blue were measured in a 5 mm injured segment beginning 5 mm distal to the aortic bifurcation, and the percentage areas were calculated using ImagePro Plus 7.0 software (Media Cybernetics, Bethesda, MD).

**Running wheel analysis**. Mice were individually housed and had free access to a running wheel. Custom-built hardware and software allowed detailed character-istics of running activity to be recorded for each animal. A mouse was considered to be active when there were $\geq 2$ revolutions of the running wheel during each 1 min recording period. This equates to ≈10% of the mean dark cycle velocity of running for control animals (0.34 m/s). Continuous periods of activity (bouts) were defined as activity seen in two or more consecutive minutes. One Piezo1$^{\Delta EC}$ mouse showed complete inactivity for the first 3 days and was excluded from the analysis along with its control genotype pair.

**Cell and tissue staining**. Epididymal fat pad and liver tissues were fixed for 48 h in 4% PFA at 4 °C prior to processing on a Leica ASP 200 and embedding in CellWax (Cellpath) on a Leica EG1150H embedding station. Sections of 4 μm were cut on a Leica RM2235 microtome onto Plus Frost slides (Solmedia) and allowed to dry at 37 °C overnight prior to staining. Slides were de-waxed in xylene and rehydrated in

ethanol. H&E was performed by staining in Mayer's Haematoxylin for 2 min and eosin for 2 min. Slides were imaged on an Aperio AT2 (Leica Biosystems) high definition digital pathology slide scanner with a maximal magnification of ×20. Tissue processing and imaging were performed at Section of Pathology and Tumour Biology, Leeds Institute of Cancer and Pathology. For CD31 immunofluorescence, cells were fixed on coverslips in 4% paraformaldehyde and permeablised with 0.1% TritonX-100 at room temperature. Cells were blocked with donkey serum for 30 min to prevent non-specific binding. Cells were then incubated with 1% BSA in PBS containing rabbit anti-mouse CD31 (1:50, Abcam ab28364). Following incubation with primary antibody, cells were washed in PBS and incubated with Alexa Fluor 488-conjugated affinipure donkey anti-rabbit IgG (1:300, Jackson Immuno Research Laboratories) for 45 min at room temperature. Cells were mounted with Prolong Gold Antifade Reagent containing DAPI (Invitrogen) and visualised using a LSM 880 confocal microscope (Zeiss).

**RNA isolation and quantitative PCR**. Total RNA was isolated using a standard TriReagent protocol and treated with DNAse (TURBO DNA-free, AM1907M, Ambion). An aliquot was used for cDNA synthesis using a High Capacity RNA-to-cDNA kit (Applied Biosystems, UK) containing Oligo-dT and random primers. Real-time PCR was performed using Roche Fast Start SYBR Green I on a Lightcycler2 with Lightcycler 3.5 software or using Roche 480 SYBR Green I on a Lightcycler480II with Lightcycler 1.5.62 software. DNA amplification was for 35 cycles with an initial 10 min at 95 °C followed by 10 s at 95 °C, 6 s at 55 °C and 14 s at 72 °C. Primers were used at 0.5 μM. Sequences of PCR primers are specified in Supplementary Table 1. The specificity of PCR was verified by reactions without RT (-RT) and by melt-curve analysis. PCR cycle crossing-points (CP) were determined by fit-points methodology. Relative abundance of target RNA was calculated from (E18sCp)/(EtargetCp). All quantitative PCR reactions were performed in duplicate and the data averaged to generate one value per experiment.

**Data analysis**. Genotypes of mice were always blinded to the experimenter and mice were studied in random order determined by the genotype of litters. Data were generated in pairs (control mice and Piezo1$^{\Delta EC}$ mice) and data sets compared statistically by independent $t$-test without assuming equal variance. Paired $t$-tests were used when comparing data before and after application of flow or a substance to the same membrane patch or cell. Statistical significance was considered to exist at probability $(P) < 0.05$ (* < 0.05, ** < 0.01, *** < 0.001). Where data comparisons lack an asterisk, they were not significantly different. The number of independent experiments (mice or independent cell cultures) is indicated by $n$. For multi-well assays or multiple cell on coverslip studies, the number of replicates is indicated by $N$. Descriptive statistics are shown as mean ± s.e.m. unless indicated as mean ± s.d. (standard deviation). Origin Pro software was used for data analysis and presentation.

**Data availability**. All relevant data are available from the authors upon reasonable request.

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

## Acknowledgements

The research was supported by grants from the Medical Research Council, Wellcome Trust and British Heart Foundation. N.E. was supported by a scholarship from the Libyan Government and P.J.W. was supported by the Leeds Teaching Hospitals Trust Charitable Foundation.

## Author contributions

B.R. coordinated experimental work and data analysis, generated and validated mice, performed experimental studies, generated the HEKT-REx-293-Piezo1 cell line, proposed ideas for experiments and wrote parts of the manuscript. J.S. generated and analysed electrophysiology and pressure myography data. N.E. generated contraction data and analysed the data with B.R. M.J.D. performed telemetry studies and analysed the data with B.R. P.J.W. and H.J.G. isolated and analysed endothelial cells. S.J.L. performed and analysed physical performance studies with B.R. and S.C.C. M.A.B. performed and analysed ultrasound studies with B.R. N.Y.Y. performed the arterial injury studies and analysed the data with B.R. M.J.L. performed and analysed the thallium flux assays. R.M.C. performed and analysed the retinal vasculature studies with B.R. J.L. generated the initial cross of Piezo1$^{flox}$ mice with mice expressing Cadh5-cre. K.M. and T.S.F. maintained mouse colonies and assisted with ethical compliance. H.V. provided technical advice for myography studies. K.C. and R.F. synthesised Yoda1. P.D.B. advised on statistical analysis. A.W. developed hardware for acquiring data and software for analysis of physical performance data. L.M., S.C.C., P.S., S.B.W. and M.T.K. provided intellectual

input. All authors commented on the manuscript. D.J.B. initiated the project, generated research funds and ideas, led and coordinated the project, interpreted data and wrote most of the paper.

## Additional information

**Competing interests:** The authors declare no competing financial interests.

