## [Peer Review File · Nature Communications]

Reviewers' comments:

Reviewer #1 (Remarks to the Author):

Review of NatComm-119830 "Specialised Piezo1 channels sense whole body physical activity to reset cardiovascular homeostasis and enhance performance"

A role for Piezo1 channels in the regulation of mesenteric vascular resistance is described, specifically in response to elevated flow rate-driven mesenteric vasoconstriction during exercise. Endothelium specific inducible Piezo1 null mice show depressed systemic blood pressure during exercise. The manuscript puts forth Piezo1-dependent inhibition of endothelial-derived hyperpolarizing factor (EDHF) as a mechanism for these changes.

The mechanisms and balance of endothelial and smooth muscle interplay to determine local and global vascular resistances is a highly complex field with many outstanding questions. The role of Piezo1 in this context is of high general interest and appropriate in scope.

The major issue with the manuscript is in relation to recently published results (Wang et al, 2016), describing a role of Piezo1 directly in opposition of what is presented here. The authors do not discuss or address this discrepancy sufficiently. Additionally, the suggested mechanistic insight relies only on indirect measures and, in many cases, conjecture without evidence is presented. Addressing these issues experimentally would greatly increase enthusiasm for acceptance of this manuscript.

SPECIFIC MAJOR COMMENTS

1. The authors do not show that their knockout strategy indeed completely ablates endothelial cells of Piezo1 expression. Since smooth muscle cells also express Piezo1, the authors should isolate endothelial cells and perform RT-QPCR to verify knockdown. (this point might end up being relevant to points 2 and 3 below).

2. Similar to the present manuscript, Wang et al. use *Cdh5-CreERT*;Piezo1 Flox mice and telemetric measurements of blood pressure in awake animals. This other manuscript comes to the conclusion that the absence of Piezo1 leads to elevated systemic blood pressure at rest. This is in apparent contradiction with the data presented here, showing a trend towards lower blood pressures at rest and strongly decreased mean arterial pressure during exercise. The reason for this should be more carefully studied and addressed both experimentally and in discussion.

3. As in point 1, Wang et al use mesenteric arteries from these animals and study flow-diameter relationship in control and Piezo1 knockout animals. Once again, the conclusion of these experiments is opposite than what one would draw based on the data presented here. Additional experiments are likely necessary to clarify this important issue. For example, the authors should attempt to recapitulate the flow-dependent arterial diameter changes recorded by Wang et al. Repeating experiments in identical experimental settings could shed light on why the two groups observe seemingly contradictory results.

4. The authors suggest a mechanism whereby Piezo1-dependent sustained depolarization in endothelial cells is transmitted (presumably via myo-endothelial junctions) smooth muscle, leading to increased contractility. This is based on the difference observed with BK channel inhibitors on isometric tension, and on measures of membrane potential changes in endothelium. The authors extrapolate the effects of such changes onto measurements made directly on smooth muscle cells. However, there is no data demonstrating the extent to which endothelial membrane potential is coupled to that of the smooth muscle layer. Is there evidence that changes in endothelial membrane potential directly translate to equivalent changes in smooth muscle cells? Additionally, how does one account for the fact that Piezo1 is in fact calcium-permeable, and is thereby expected to also activate BK channels?

MINOR COMMENTS

1. The wording of the manuscript and discussion suggest that the endothelium possesses a "special kind" of Piezo1, which does not inactivate. While it is likely true that Piezo1 is slower in these cells

than that observed via over-expression in heterologous systems, recent publications (Cox et al, 2015; Lewis et al, 2015) caution against extrapolating inactivation kinetics observed in excised patches to scenarios of intact cells.

2. The justification of the use of the FluxOR TI+ assay is unclear. Piezo1 channels are calcium-permeable, fluorescent calcium measurements have both a higher sensitivity and a wider dynamic range as compared to this assay. Furthermore, the authors utilize calcium measurements for other figures. The use of this assay should be explained.

Reviewer #2 (Remarks to the Author):

Rode et al present a truly impressive exploration of the possible roles of endothelial Piezo1 channels in the vascular function. Previous studies of the group and others showed that Piezo1 channels play a major role in the vascular development but its role in the adult endothelium has not been established. The main findings of the study are (i) a lack of endothelial Piezo1 prevented the elevation of blood pressure during physical activity; (ii) Piezo1 are expressed in mesenteric endothelial cells and its activity is sensitive to flow; (iii) flow-induced activation of Piezo1 may generate a depolarizing shift of the membrane potential that might be sufficient to activate voltage-gated Ca channels in the smooth muscle cells which would result in vasoconstriction; (iv) there is a transient decrease in mice activity in Piezo1 Δ EC mice and relatively small but significant weight loss after the exercise. Based on these findings, it is proposed that Piezo1 plays an important role in vascular homeostasis during physical activity by inducing constriction of the mesenteric arteries, elevating the blood the pressure and allowing the blood flow to redirect from intestines to skeletal muscle and possibly other organs. This is a very interesting and provocative hypothesis but the current findings do not provide compelling evidence to support this hypothesis. There are several major questions that need to be answered.

Major comments:

1. Flow-induced vasoconstriction: The key part of the hypothesis is that flow should induce vasoconstriction in mesenteric artery and that this effect should depend on the expression of endothelial Piezo1. This needs to be demonstrated. Demonstrating that increase in flow can cause vasoconstriction in mesenteric artery is particularly important not only because it is the corner stone of the proposed hypothesis but also because it is contrary to the well-established vasodilatory effect of flow demonstrated in numerous studies. Indeed, flow-induced vasodilation is typically measured in pre-constricted vessels and therefore it is not impossible that flow can also have a vasoconstriction effect under different circumstances but demonstrating this is a key validation of the proposed hypothesis.

2. Piezo1 channels as flow sensors: The authors clearly show an increase in the channel activity in response to a physical force but there are three concerns about these experiments:

a. One concern is methodological. The recordings are performed in the outside-out configuration when a recording pipette with a membrane patch is inserted into a capillary tube. A concern for this setup is that flow experienced by a cell or membrane inserted into a tube by a recording pipette is very different in nature from a fluid shear stress experienced by a cell attached to a substrate. Inserting a cell into a tube would generate a force more similar to hitting it with a fluid stream like a jet. It is possible to overcome this problem by seeding cells into the tube and then inserting a recording pipette and establishing a tight seal to a substrate-attached cell near the tube opening.

b. Another related concern is the outside-out configuration. When excised patch is created, either outside-out or inside-out, the membrane patch tends actually to move to the inside of the pipette tip. It is still exposed to the bath solutions but may actually be protected from the flow. In this sense, it is really not clear what the channels are responding to. Again, whole cell recording as suggested above would be more convincing in terms of demonstrating the flow-sensitivity of the channels. Indeed, in this case, it would be impossible to say whether other cellular components contribute to shear-stress

induced activation of the Piezo1 but it would provide a firm evidence for their shear-stress sensitivity in these cells.

c. The range of shear stresses used in this study is very high, somewhere between 50 and 180 dyn/cm², the more standard range is 10-20 dyn/cm² and there is some literature reporting 7 to 50 dynes/cm² in small arteries. Is there a justification to use shear stresses so much above this range? If the current shear stress range is significantly above the physiological range, it is a significant concern.

3. Piezo1-dependent depolarization of vascular smooth muscle cells: It is definitely possible that depolarization of endothelial cells could propagate through tight junction to the smooth muscle cells resulting in their depolarization. A similar mechanism is believed to be responsible for the endothelial hyperpolarizing factor. However, there are also the following concerns:

a. Quantitative analysis of the threshold presented in Figure 3 seems to assume that a voltage shift in VSMCs should match the voltage shift in ECs, which is very unlikely to be the case. In contrast, it is expected that there will be significant dissipation. It is not clear, therefore, whether indeed Piezo can generate enough depolarization in ECs in response to flow to activate voltage-gated Ca channels in VSMCs. The ideal experiment would be to have a co-culture of ECs with VSMCs and show VSMC depolarization or Ca response to flow that would disappear when VSMCs are co-cultured with Piezo1 Δ EC cells. Alternatively, it is possible to show that flow induces a Ca response in VSMCs in mesenteric arteries of control mice but not in Piezo1 Δ EC mice.

b. The range of shear stresses needed to generate a sufficient voltage shift in VSMCs even not taking into the account the dissipation effect is very high > 100 dyn/cm². The authors should provide some evidence that this range is physiological.

4. Physical performance depends on endothelial Piezo1. The weight loss in exercised Piezo EC KO mice is very interesting, however, the performance data in Figure 4 building up to this weight loss is not as compelling. For both, the distance run on the wheel and percentage time for which mice were active on the wheel show differences between control and Piezo1 Δ EC mice only on day one after introducing the wheel after which the difference disappears. It is not clear what is the meaning and significance of this transient nature of the Piezo effects. To compare exercise capacity/fatigue between WT and Piezo EC KO mice, it might be better to perform acute bouts of forced exercise to exhaustion.

Irena Levitan, PhD
Professor of Medicine, Pharmacology and Bioengineering
University of Illinois at Chicago

Reviewer #3 (Remarks to the Author):

The authors performed an extensive work to show that the endothelial piezo1 channel has a moderate role in basal conditions in mesenteric arteries. Nevertheless, when stimulated by exercising, these receptors seems to activated endothelium-dependent transfer of vasoconstrictors signaling to smooth muscle cells. The study is done properly using appropriate approaches. The data support the conclusion and the discussion is clear enough with reasonable arguments. Nevertheless, major concerns remain, especially as regard to Piezo1-dependent intercellular communication.

Major comments:

1- The interaction between endothelial cells and smooth muscle cells allowing piezo1-dependent transfer of vasoconstriction from endothelial cells to smooth muscle is probably a property of the mesenteric circulation as explained by obvious physiological reasons. To strengthen the hypothesis, the author should show that this does not happen in skeletal muscle arteries where a bulk vasodilator effect should be evidenced; eventually in piezo1-dependent fashion.

2- A possible explanation could be a pH effect on the channel activity and/or on the interaction between endothelial cells and smooth muscle cells when mice are submitted to physical activity.

3- The investigation of endothelium-dependent dilatation should induce a study of flow-mediated dilatation as in Wang et al. (J Clin Invest. 2016;126(12):4527-4536). These authors observed a strong réduction in FMD in mesenteric arteries isolated from mice lacking endothelial piezo1. Similarly, as the mechanism described by Wang et al involves a piezo1-dependent release of ATP, which will activate P2Y2 receptor, a pH effects remain highly possible. ATP production and release is closely related to H⁺ metabolism.

NCOMMS-17-01887-T

Revised Manuscript

Rode et al “Piezo1 channels sense whole body physical activity to reset cardiovascular homeostasis and enhance performance”

Responses to Reviewer #1

Thank you for the important feedback which we very much appreciate. In response we performed more experiments and revised the manuscript accordingly. We also provide more discussion of key issues and better contextualization of the findings as well as generally raising the quality of the work and its presentation. A red-highlighted version of the manuscript is included to make it clear where changes have been made.

“Reviewer #1 (Remarks to the Author):”

“Review of NatComm-119830 “Specialised Piezo1 channels sense whole body physical activity to reset cardiovascular homeostasis and enhance performance”

A role for Piezo1 channels in the regulation of mesenteric vascular resistance is described, specifically in response to elevated flow rate-driven mesenteric vasoconstriction during exercise. Endothelium specific inducible Piezo1 null mice show depressed systemic blood pressure during exercise. The manuscript puts forth Piezo1-dependent inhibition of endothelial-derived hyperpolarizing factor (EDHF) as a mechanism for these changes.

The mechanisms and balance of endothelial and smooth muscle interplay to determine local and global vascular resistances is a highly complex field with many outstanding questions. The role of Piezo1 in this context is of high general interest and appropriate in scope.

The major issue with the manuscript is in relation to recently published results (Wang et al, 2016), describing a role of Piezo1 directly in opposition of what is presented here. The authors do not discuss or address this discrepancy sufficiently.”

We added more discussion of the Wang et al study along with new myography data to help understanding of our work in relation to this prior work (Figs. 6 & 8).

Wang et al 2016 did not address the question of whole body physical activity, so their work does not overlap with the main concept of our study.

Although Wang et al did not include the data in their publication, they subsequently showed us that diastolic blood pressure was not changed in their experiments. We also observed no change in diastolic pressure in non-exercising mice (Fig. 3a), so in this regard our results are similar. In apparent contrast to Wang et al we did not observe increased systolic pressure in non-exercising mice (Fig. 3a), but there were differences in how our studies were performed: for example, Wang et al combined data from male and female mice (we only used males), Wang et al did not provide access to a running wheel (we did, so our non-exercising periods were between periods of exercise) and Wang et al used the Tie2 promoter for endothelial deletion (we used the Cadh5 promoter). Given the absence of any data from Wang et al on heart rate or other cardiac parameters, an off-target action of their Tie2 promoter cannot be excluded as a potential explanation for the elevated systolic pressure. Alternatively the systolic effect might be a feature of female mice, or there could be other explanations. We provide more extensive phenotyping of our mice (Fig. 1).

We believe that a key factor in the apparent discrepancies is the underlying biology: the dichotomy presented by Piezo1 channels – channels which can drive Ca^{2+} -activation and membrane depolarisation, which in the endothelium are, or can be, in opposition. We address this important point in the manuscript and have sought to clarify it further by more text and a schematic (Fig. 2c). That is, there is not necessarily disagreement between us and Wang et al. It may be that the biology expresses itself differently depending on the conditions. We discuss this idea in the manuscript.

“Additionally, the suggested mechanistic insight relies only on indirect measures and, in many cases, conjecture without evidence is presented. Addressing these issues experimentally would greatly increase enthusiasm for acceptance of this manuscript.

We added new myography data showing flow-induced vasoconstriction in mesenteric artery and its inhibition by a calcium antagonist; i.e. blockade of voltage-gated Ca^{2+} channels (Fig. 6). These experimental data provide direct evidence supporting our hypothesis that flow stimulates Piezo1 in the endothelium to drive depolarization of the endothelium which is coupled electrically to the underlying vascular smooth muscle cells and activation of voltage-gated Ca^{2+} channels in the smooth muscle cells.

We added new data showing that saphenous and carotid arteries lack the Piezo1-dependence of mesenteric artery (Fig. 8). These experimental data provide direct evidence supporting our hypothesis that whole body physical activity stimulates Piezo1 channels in the mesenteric arteries to constrict these arteries but not skeletal muscle or cerebral arteries, thus redistributing blood flow from the GI tract to skeletal muscle and the brain.

“SPECIFIC MAJOR COMMENTS”

“1. The authors do not show that their knockout strategy indeed completely ablates endothelial cells of Piezo1 expression. Since smooth muscle cells also express Piezo1, the authors should isolate endothelial cells and perform RT-QPCR to verify knockdown. (this point might end up being relevant to points 2 and 3 below).”

We isolated endothelial cells and showed that the knockout strategy selectively and completely ablates responses to the Piezo1 agonist, Yoda1, and that there is strong mRNA depletion (Supplementary Figure 2).

“2. Similar to the present manuscript, Wang et al. use *Cdh5-CreERT;Piezo1 Flox* mice and telemetric measurements of blood pressure in awake animals. This other manuscript comes to the conclusion that the absence of Piezo1 leads to elevated systemic blood pressure at rest. This is in apparent contradiction with the data presented here, showing a trend towards lower blood pressures at rest and strongly decreased mean arterial pressure during exercise. The reason for this should be more carefully studied and addressed both experimentally and in discussion.”

We have addressed this topic as described above.

“3. As in point 1, Wang et al use mesenteric arteries from these animals and study flow-diameter relationship in control and Piezo1 knockout animals. Once again, the conclusion of these experiments is opposite than what one would draw based on the data presented here. Additional experiments are likely necessary to clarify this important issue. For example, the authors should attempt to recapitulate the flow-dependent arterial diameter changes recorded

by Wang et al. Repeating experiments in identical experimental settings could shed light on why the two groups observe seemingly contradictory results.”

We have addressed this topic as described above.

Flow-induced vasoconstriction is evident in the data of Wang et al (their Fig. 5A).

“4. The authors suggest a mechanism whereby Piezo1-dependent sustained depolarization in endothelial cells is transmitted (presumably via myo-endothelial junctions) smooth muscle, leading to increased contractility. This is based on the difference observed with BK channel inhibitors on isometric tension, and on measures of membrane potential changes in endothelium. The authors extrapolate the effects of such changes onto measurements made directly on smooth muscle cells. However, there is no data demonstrating the extent to which endothelial membrane potential is coupled to that of the smooth muscle layer. Is there evidence that changes in endothelial membrane potential directly translate to equivalent changes in smooth muscle cells? Additionally, how does one account for the fact that Piezo1 is in fact calcium-permeable, and is thereby expected to also activate BK channels?”

Elegant prior work demonstrated that voltage shifts in the smooth muscle layer of mesenteric artery closely match those in the endothelium (Yamamoto et al 1999 *J Physiol* 514, 505-513) [Ref. 18].

The concept of efficient electrical coupling between the endothelium and smooth muscle layer is well established and central to the well-accepted EDH(F) mechanism (Garland & Dora 2016 *Acta Physiol* doi:10.1111/apha.12649) [Ref. 10]. Although the efficiency and relevance of EDH(F) varies in different arteries, it is an established phenomenon in mouse mesenteric arteries (Boedtkjer et al 2013 *J Physiol* 591, 1447-1461) [Ref. 19]. We propose our Piezo1 mechanism as an “anti-EDH(F)” and this is nicely illustrated by our data of Fig. 2d, e.

The new myography data which we have added demonstrate that flow against the endothelial layer is sufficient to activate voltage-gated Ca^{2+} channels in the underlying smooth muscle layer (Fig. 6).

We agree that the question about Ca^{2+} -activated K^+ channels and the many other Ca^{2+} -activated or Ca^{2+} -facilitated mechanisms is interesting. We describe this as the dichotomy presented by Piezo1 channels in the endothelium – channels which can drive Ca^{2+} -activation and membrane depolarisation, which in the endothelium are, or can be, in opposition. We address this important point in the manuscript and have sought to clarify it further by more text and a schematic (Fig. 2c). We are interested in how this dichotomy might be controlled. One possibility is through spatial localisation of Piezo1 relative to other mechanisms.

“MINOR COMMENTS”

“1. The wording of the manuscript and discussion suggest that the endothelium possesses a “special kind” of Piezo1, which does not inactivate. While it is likely true that Piezo1 is slower in these cells than that observed via over-expression in heterologous systems, recent publications (Cox et al, 2015; Lewis et al, 2015) caution against extrapolating inactivation kinetics observed in excised patches to scenarios of intact cells.”

We found the non-inactivating property in intact endothelial cell recordings (Fig. 4) as well as excised patch recordings (Fig. 3).

Prior work has suggested the possibility of a non-inactivating state but particular stimuli were applied to achieve the effect and the property is not usually associated with Piezo1 channels [Refs. 16, 17]. The dominant finding reported in the literature is that Piezo1 channels are very rapidly activating and rapidly inactivating. Here we show endothelial Piezo1 channels which intrinsically lack inactivation.

We deleted “Specialised” from the title but think it is important to retain emphasis on the non-inactivating feature as it is presumably crucial to achieving the sustained depolarization which is necessary for the proposed mechanism. We hope this revised emphasis is acceptable.

“2. The justification of the use of the FluxOR TI^+ assay is unclear. Piezo1 channels are calcium-permeable, fluorescent calcium measurements have both a higher sensitivity and a wider dynamic range as compared to this assay. Furthermore, the authors utilize calcium measurements for other figures. The use of this assay should be explained.”

The data are moved to Supplementary Fig. 6. The assay enables measurement of unidirectional ion flux because TI^+ is physiologically absent from the cells yet permeant in these channels. Therefore addition of TI^+ to the extracellular medium allows an assessment of its influx and thus the constitutive channel activity. Interpretation of Ca^{2+} studies is more complex because of the cell’s well-developed Ca^{2+} influx, efflux and storage mechanisms.

Responses to Reviewer #2

Thank you for the important feedback, which we very much appreciate. In response we performed more experiments and revised the manuscript accordingly. We also provide more discussion of key issues and better contextualization of the findings as well as generally raising the quality of the work and its presentation. A red-highlighted version of the manuscript is included to make it clear where changes have been made.

“Reviewer #2 (Remarks to the Author):”

“Rode et al present a truly impressive exploration of the possible roles of endothelial Piezo1 channels in the vascular function. Previous studies of the group and others showed that Piezo1 channels play a major role in the vascular development but its role in the adult endothelium has not been established. The main findings of the study are (i) a lack of endothelial Piezo1 prevented the elevation of blood pressure during physical activity; (ii) Piezo1 are expressed in mesenteric endothelial cells and its activity is sensitive to flow; (iii) flow-induced activation of Piezo1 may generate a depolarizing shift of the membrane potential that might be sufficient to activate voltage-gated Ca channels in the smooth muscle cells which would result in vasoconstriction; (iv) there is a transient decrease in mice activity in Piezo1 Δ EC mice and relatively small but significant weight loss after the exercise. Based on these findings, it is proposed that Piezo1 plays an important role in vascular homeostasis during physical activity by inducing constriction of the mesenteric arteries, elevating the blood the pressure and allowing the blood flow to redirect from intestines to skeletal muscle and possibly other organs. This is a very interesting and provocative hypothesis but the current findings do not provide compelling evidence to support this

hypothesis. There are several major questions that need to be answered.”

“Major comments:”

“1. Flow-induced vasoconstriction: The key part of the hypothesis is that flow should induce vasoconstriction in mesenteric artery and that this effect should depend on the expression of endothelial Piezo1. This needs to be demonstrated. Demonstrating that increase in flow can cause vasoconstriction in mesenteric artery is particularly important not only because it is the corner stone of the proposed hypothesis but also because it is contrary to the well-established vasodilatory effect of flow demonstrated in numerous studies. Indeed, flow-induced vasodilation is typically measured in pre-constricted vessels and therefore it is not impossible that flow can also have a vasoconstriction effect under different circumstances but demonstrating this is a key validation of the proposed hypothesis.”

Thank you for your understanding regarding the possibility of vasoconstriction.

We added new pressure myography data showing flow-induced vasoconstriction in mesenteric artery and its inhibition by calcium antagonist i.e. L-type voltage-gated Ca^{2+} channel blockade (Fig. 6). These data support our hypothesis that flow stimulates Piezo1 in the endothelium to drive depolarization of the endothelium which is coupled electrically to the underlying vascular smooth muscle cells and sufficient for activation of voltage-gated Ca^{2+} channels in the smooth muscle cells. We show that without Piezo1 there is no flow-induced depolarization (Fig. 5a) and that exercised-induced elevation of blood pressure is blunted (Fig. 3b).

We hypothesise that a key factor in the balance between vasodilator and vasoconstrictor mechanisms is the dichotomy presented by Piezo1 channels in the endothelium – channels which can drive Ca^{2+} -activation (e.g. of eNOS) and membrane depolarisation. We address this important point in the manuscript and have sought to further clarify it through more text and a schematic (Fig. 2c).

In our new studies showing flow-induced vasoconstriction we applied flow in the presence of L-NAME, which suppressed nitric oxide production and flow-mediated vasodilatation (Fig. 6). We have added focussed discussion of this topic.

We do not doubt the existence of flow-mediated vasodilatation or its importance. Our point is that flow-mediated vasoconstriction is also possible and that it mediates mesenteric vasoconstriction in whole body physical exercise, directing blood away from the GI tract. Such vasoconstriction is a recognised and important phenomenon in whole body physical activity. We revised the manuscript to clarify these points.

Our blood pressure measurements are consistent with Piezo1 mediating a vasoconstrictor effect because exercise-induced elevation of blood pressure was blunted in the Piezo1 knockout mice (Fig. 3b). We are therefore suggesting that the vasoconstrictor effect of endothelial Piezo1 is dominant in vivo in healthy mice when they are exercising. This implies that nitric oxide-mediated vasodilatation of *mesenteric* arteries is relatively less important in this whole body exercise situation. This does not mean that nitric oxide-mediated vasodilatation is relatively unimportant in mesenteric arteries of mice in all other situations or other vascular beds. We have added discussion of these points.

Consistent with our ideas, flow-induced vasoconstriction is evident in the mesenteric artery studies of Wang et al (*JCI* 2016 126, 4527-4536) – vasodilatation occurred at low flow but vasoconstriction at higher flow (their Fig. 5A). These were *in vitro* studies. In *exercise* (*in vivo*) we are suggesting that vasoconstriction is relatively more important in *mesenteric* artery (Fig. 3b).

The agent used to induce pre-constriction is important in *in vitro* myograph studies and may or may not mimic the *in vivo* situation. *In vivo*, myogenic tone may be dominant and agonist-induced tone relatively less important.

“2. Piezo1 channels as flow sensors: The authors clearly show an increase in the channel activity in response to a physical force but there are three concerns about these experiments: a. One concern is methodological. The recordings are performed in the outside-out configuration when a recording pipette with a membrane patch is inserted into a capillary tube. A concern for this setup is that flow experienced by a cell or membrane inserted into a tube by a recording pipette is very different in nature from a fluid shear stress experienced by a cell attached to a substrate. Inserting a cell into a tube would generate a force more similar to hitting it with a fluid stream like a jet. It is possible to overcome this problem by seeding cells into the tube and then inserting a recording pipette and establishing a tight seal to a substrate-attached cell near the tube opening.”

An important technical principle of our work has been the study of freshly-isolated endothelium to avoid unknown and potentially non-physiological changes associated with cell culture and adhesion of cells to foreign substrates. Piezo1 expression and function is particularly susceptible to the nature of the substrate (Pathak *et al* 2014 *PNAS* 111, 16148-53). Therefore we are concerned that culturing cells in a tube would distort the role of Piezo1.

We made observations in outside-out patches (Fig. 4) but also similar observations by recording membrane potential from multicellular endothelial fragments (Fig. 5).

We further addressed the physiological relevance by adding the flow-induced vasoconstriction data obtained from intact artery where fluid flow was applied to endothelium attached to its normal substrate (Fig. 6). Our blood pressure data also suggest physiological relevance of the vasoconstriction (Fig. 3b).

“b. Another related concern is the outside-out configuration. When excised patch is created, either outside-out or inside-out, the membrane patch tends actually to move to the inside of the pipette tip. It is still exposed to the bath solutions but may actually be protected from the flow. In this sense, it is really not clear what the channels are responding to. Again, whole cell recording as suggested above would be more convincing in terms of demonstrating the flow-sensitivity of the channels. Indeed, in this case, it would be impossible to say whether other cellular components contribute to shear-stress induced activation of the Piezo1 but it would provide a firm evidence for their shear-stress sensitivity in these cells.”

Outside out patches remain outside the pipette tip (Lewis & Grandl 2015 *Elife* 4. pii: e12088. doi: 10.7554/eLife.12088) and so channels in outside-out patches directly experienced the fluid flow. We added text on this point and a reference [Ref. 11].

We also provide similar data for intact endothelial cell recordings (Fig. 5). Our observations for excised patches and whole cells were similar.

“c. The range of shear stresses used in this study is very high, somewhere between 50 and 180 dyn/cm², the more standard range is 10-20 dyn/cm² and there is some literature reporting 7 to 50 dynes/cm² in small arteries. Is there a justification to use shear stresses so much above this range? If the current shear stress range is significantly above the physiological range, it is a significant concern.”

The use of 10-20 dyn/cm² (1-2 Pa) is quite common in in vitro studies but not necessarily a reflection of what happens in physiology (Cheng et al 2007 *Atherosclerosis* 195, 225–235) [Ref. 23]. Shear stress values vary substantially depending on vascular calibre, vascular architecture and species. In mice, shear stress can be at least 10 times more than in humans. The values we used were appropriate to the context and we have added text to clarify this point: “Incremental increases in fluid flow up to and above fluid flow rates reported in anaesthetised mice (Ref. 15)”; “shear stress in mice is considered to range from about 3 to 60 Pa (Refs. 22-24), consistent with our studies (Fig. 5d). Values in humans are usually lower (Refs. 22, 23)”.

“3. Piezo1-dependent depolarization of vascular smooth muscle cells: It is definitely possible that depolarization of endothelial cells could propagate through tight junction to the smooth muscle cells resulting in their depolarization. A similar mechanism is believed to be responsible for the endothelial hyperpolarizing factor. However, there are also the following concerns: a. Quantitative analysis of the threshold presented in Figure 3 seems to assume that a voltage shift in VSMCs should match the voltage shift in ECs, which is very unlikely to be the case. In contrast, it is expected that there will be significant dissipation. It is not clear, therefore, whether indeed Piezo can generate enough depolarization in ECs in response to flow to activate voltage-gated Ca channels in VSMCs. The ideal experiment would be to have a co-culture of ECs with VSMCs and show VSMC depolarization or Ca response to flow that would disappear when VSMCs are co-cultured with Piezo1 Δ EC cells. Alternatively, it is possible to show that flow induces a Ca response in VSMCs in mesenteric arteries of control mice but not in Piezo1 Δ EC mice.”

Elegant prior work demonstrated that voltage shifts in mesenteric VSMCs closely match those in ECs (Yamamoto et al 1999 *J Physiol* 514, 505-513) [Ref. 18]. The concept of efficient electrical coupling between the endothelium and smooth muscle layer is established and central to the accepted EDH(F) mechanism (Garland & Dora 2016 *Acta Physiol* doi:10.1111/apha.12649) [Ref. 10]. Although the efficiency and relevance of EDH(F) varies in different arteries, its existence is demonstrated in mouse mesenteric arteries (Boedtkjer et al 2013 *J Physiol* 591, 1447-1461) [Ref. 19]. We propose our Piezo1 mechanism as an “anti-EDH(F)” and provide evidence for this through the data of Fig. 2d, e.

To further test our ideas we have performed new myography experiments, the results of which demonstrate existence of flow-induced vasoconstriction via coupling of the endothelium to vascular smooth muscle cells and their voltage-gated Ca²⁺ channels (Fig. 6).

Studies of co-cultured cells are unlikely to satisfactorily address this biology because there would be concerns about whether the cultured cells retained the appropriate physiological phenotype (e.g. L-type Ca²⁺ channel expression in VSMCs) or achieved the necessary micro-anatomy (e.g. myo-endothelial junction structures).

“b. The range of shear stresses needed to generate a sufficient voltage shift in VSMCs even not

taking into the account the dissipation effect is very high $> 100 \text{ dyn/cm}^2$. The authors should provide some evidence that this range is physiological.”

We have addressed this topic as described above.

“4. Physical performance depends on endothelial Piezo1. The weight loss in exercised Piezo EC KO mice is very interesting, however, the performance data in Figure 4 building up to this weigh loss is not as compelling. For both, the distance run on the wheel and percentage time for which mice were active on the wheel show differences between control and Piezo1 Δ EC mice only on day one after introducing the wheel after which the difference disappears. It is not clear what is the meaning and significance of this transient nature of the Piezo effects. To compare exercise capacity/fatigue between WT and Piezo EC KO mice, it might be better to perform acute bouts of forced exercise to exhaustion.”

The data for the first night on the wheel reflect the performance of the animals without prior training (i.e. when they were relatively unfit). It is correct that the effect of Piezo1 was on this first night and to a less extent the second night (i.e. the effect was lost with training). We think the training enabled compensation as the animals became fitter. The weight loss data nevertheless suggest that the animals were working harder. We have included discussion of these points.

Responses to Reviewer #3

Thank you for the important feedback, which we very much appreciate. In response we performed more experiments and revised the manuscript accordingly. We also provide more discussion of key issues and better contextualization of the findings as well as generally raising the quality of the work and its presentation. A red-highlighted version of the manuscript is included to make it clear where changes have been made.

“Reviewer #3 (Remarks to the Author):”

“The authors performed an extensive work to show that the endothelial piezo1 channel has a moderate role in basal conditions in mesenteric arteries. Nevertheless, when stimulated by exercising, these receptors seems to activated endothelium-dependent transfer of vasoconstrictors signaling to smooth muscle cells. The study is done properly using appropriate approaches. The data support the conclusion and the discussion is clear enough with reasonable arguments. Nevertheless, major concerns remain, especially as regard to Piezo1-dependent intercellular communication.”

“Major comments:”

“1- The interaction between endothelial cells and smooth muscle cells allowing piezo1-dependent transfer of vasoconstriction from endothelial cells to smooth muscle is probably a property of the mesenteric circulation as explained by obvious physiological reasons. To strengthen the hypothesis, the author should show that this does not happen in skeletal muscle arteries where a bulk vasodilator effect should be evidenced; eventually in piezo1-dependent fashion.”

We have added contraction study data for saphenous artery and carotid artery to provide information on the relevance to skeletal muscle and cerebral vessels. In both cases the role of Piezo1 which we show in mesenteric artery is lacking (Fig. 8). These data are consistent with the fact that blood flow to the GI tract is reduced in whole body physical activity but increased or maintained in skeletal muscle and brain. We have included a new section on the vascular bed-specific role of the Piezo1.

“2- A possible explanation could be a pH effect on the channel activity and/or on the interaction between endothelial cells and smooth muscle cells when mice are submitted to physical activity.”

Acidification inhibits Piezo1 channels with a pK of 6.9 (Bae et al 2015 *JBC* 290, 5167-5173) but we have not developed this topic because of the new data suggesting no role for endothelial Piezo1 in saphenous artery (Fig. 8).

“3- The investigation of endothelium-dependent dilatation should induce a study of flow-mediated dilatation as in Wang et al. (*J Clin Invest.* 2016;126(12):4527-4536). These authors observed a strong reduction in FMD in mesenteric arteries isolated from mice lacking endothelial piezo1. Similarly, as the mechanism described by Wang et al involves a piezo1-dependent release of ATP, which will activate P2Y2 receptor, a pH effects remain highly possible. ATP production and release is closely related to H^+ metabolism.”

We added new myography data showing flow-induced vasoconstriction in mesenteric artery and its inhibition by a calcium antagonist; i.e. by L-type voltage-gated Ca^{2+} channel blockade (Fig. 8). These data support our hypothesis that flow stimulates Piezo1 in the endothelium to drive depolarization of the endothelium which is coupled electrically to the underlying vascular smooth muscle cells and activation of voltage-gated Ca^{2+} channels in the smooth muscle cells.

We believe that a key factor in the balance between vasodilator and vasoconstrictor mechanisms is the dichotomy presented by Piezo1 channels in the endothelium – channels which can drive Ca^{2+} -activation (e.g. of eNOS) and membrane depolarisation. We address this important point in the manuscript and have sought to further clarify it through more text and a schematic (Fig. 2c).

In our new studies showing flow-induced vasoconstriction we applied flow in the presence of L-NAME which induced pre-constriction on top of any spontaneous tone and suppressed nitric oxide production and flow-mediated vasodilatation (Fig. 6). We discuss the implications of this in the manuscript.

We do not doubt flow-mediation vasodilatation or its importance – its existence is demonstrated in many studies, including those of Wang et al. Our point is that flow-mediated vasoconstriction is also possible and that it mediates mesenteric vasoconstriction in whole body physical exercise, directing blood away from the GI tract. Such vasoconstriction is a recognised and important phenomenon in whole body physical activity. We revised the manuscript to clarify and discuss this point.

Our blood pressure measurements are consistent with Piezo1 mediating a vasoconstrictor effect because exercise-induced elevation of blood pressure was blunted in the Piezo1 knockout mice (Fig. 3b). We are therefore suggesting that the vasoconstrictor effect of endothelial Piezo1 is dominant in vivo in healthy mice when they are exercising. This suggests that nitric oxide-

mediated vasodilatation of *mesenteric* arteries is relatively less important in this situation of whole body exercise. This does not mean that nitric oxide-mediated vasodilatation is unimportant in mesenteric arteries of mice in all other situations or other vascular beds. We discuss this point in the revised manuscript.

Consistent with our ideas, flow-induced vasodilatation and vasoconstriction are evident in the mesenteric artery studies of Wang et al (*JCI* 2016 126, 4527-4536) – vasodilatation at low flow and vasoconstriction at higher flow (their Fig. 5A). These were *in vitro* studies. In *exercise* (*in vivo*) we are suggesting that the vasoconstriction is relatively most important in *mesenteric* artery.

The agent used to induce pre-constriction is important in *in vitro* myograph studies as it may or may not mimic the *in vivo* situation. *In vivo*, myogenic tone may be dominant and agonist-induced tone relatively less important. In the thorough recent study by Ahn et al 2016, flow-induced vasodilatation was nicely observed in murine mesenteric arteries pre-constricted with endothelin-1 (*J Physiol.* doi: 10.1113/JP273255). However, endothelin-1 inhibits voltage-gated Ca²⁺ channels (Guibert & Beech 1999 *J Physiol.* 514, 843-856). Therefore the flow-induced vasoconstriction of the type we describe (i.e. requiring voltage-gated Ca²⁺ channels) would likely be suppressed in an endothelin-1 pre-constriction experimental design.

Contrary to the findings of Wang et al, we find that depletion of Piezo1 increases flow-induced ATP release from human umbilical vein endothelial cells (unpublished data).

Reviewers' comments:

Reviewer #1 (Remarks to the Author):

Second review of NCOMMS-17-01887A

The authors have made significant progress with the manuscript, attempting to address reviewers' comments by both citing relevant literature, clarifying the manuscript language and conducting additional experiments. My recommendation for the publication of this manuscript is now fully contingent on the authors conducting the necessary control experiments to corroborate their new data included in this manuscript. Given all the proper controls, the findings of the work are relevant and should be of interest to the readership.

MAJOR POINTS

1. Figure 6. This is an important addition to the manuscript in response to reviewer requests. However, additional experiments are necessary in order for it to truly demonstrate what the authors claim, i.e. that flow-induced vasoconstriction happens in the mesentery and that this process is Piezo1 dependent. The following experiments are crucial for this figure to actually demonstrate this:

a) The stimulus here is pressure, which is problematic due to potential involvement of myogenic tone, which may make the pressure/flow relationship non-linear. The relevant variable to quantify and show should be flow, as the authors are trying to measure flow-induced vasoconstriction. For the very same reason a crucial control here is whether this vasoconstriction is endothelium-dependent: the authors must repeat this experiment after endothelium ablation to demonstrate that the vasoconstriction is indeed endothelium/flow-dependent.

b) Additionally, repeating this experiment with Piezo1-knockout mesenteric vessels would mechanistically greatly corroborate the blood pressure findings and the hypothesis put forth by the authors.

2. The authors include figure 8 to demonstrate that the mesenteric resistance arterioles are unique in displaying Piezo1-dependent inhibition of endothelial-derived hyperpolarization. However, the comparison is made with major conductance arteries whose anatomy is starkly different from the 2nd order resistance arterioles tested in the mesentery. To actually make this point, the authors should strive to make this comparison between similar order arterioles from other organs, such as skeletal muscle.

MINOR POINTS

Figure 7 raises more concerns than it actually answers, and in my opinion could be omitted from the manuscript. The explanation that the knockout mice work harder to on the treadmill is not substantiated by the ~2% change in bodyweight, which seems to be small enough to be well within the range of measurement error.

In response to the authors rebuttal, the thallium-influx experiment indeed lacks the potentially confounding effect of release from intracellular stores, which a calcium assay has. However, thallium does cross essentially all potassium channels, of which there are plenty, and this assay can therefore be confounded by contribution to the observed signal from all manner of potassium channels.

Additionally, calcium-dyes are much more sensitive. This is a minor point and does not take away from the value of the manuscript.

Reviewer #2 (Remarks to the Author):

The authors thoroughly addressed all the previous concerns and the manuscript is now acceptable for publication

Reviewer #3 (Remarks to the Author):

no further comment

NCOMMS-17-01887A

Revised Manuscript

Rode et al “Piezo1 channels sense whole body physical activity to reset cardiovascular homeostasis and enhance performance”

Responses to Reviewer #1

Thank you for the additional important feedback. In response we performed more experiments and revised the manuscript accordingly. A red-highlighted version of the manuscript is included to make it clear where changes have been made.

“The authors have made significant progress with the manuscript, attempting to address reviewers’ comments by both citing relevant literature, clarifying the manuscript language and conducting additional experiments. My recommendation for the publication of this manuscript is now fully contingent on the authors conducting the necessary control experiments to corroborate their new data included in this manuscript. Given all the proper controls, the findings of the work are relevant and should be of interest to the readership.”

Please see below.

“MAJOR POINTS

1. Figure 6. This is an important addition to the manuscript in response to reviewer requests. However, additional experiments are necessary in order for it to truly demonstrate what the authors claim, i.e. that flow-induced vasoconstriction happens in the mesentery and that this process is Piezo1 dependent. The following experiments are crucial for this figure to actually demonstrate this:

a) The stimulus here is pressure, which is problematic due to potential involvement of myogenic tone, which may make the pressure/flow relationship non-linear. The relevant variable to quantify and show should be flow, as the authors are trying to measure flow-induced vasoconstriction.”

The pressure gradient was changed, not the absolute pressure experienced by the artery (which was kept constant). This approach was used to create flow without change in absolute pressure. This approach and the nomenclature are similar to those described by Ahn et al 2016, J Physiol. doi: 10.1113/JP273255.

In all cases we demonstrated that the arteries were endothelium intact (at least 80% dilation to acetylcholine). Therefore the fluid flow must have been experienced by the endothelial cells and not the smooth muscle cells.

That we were not accidentally studying myogenic tone is shown conclusively by the new data described below (and included in the revised manuscript).

“For the very same reason a crucial control here is whether this vasoconstriction is endothelium-dependent: the authors must repeat this experiment after endothelium ablation to demonstrate that the vasoconstriction is indeed endothelium/flow-dependent.”

We have addressed this point genetically. The arising data show that endothelium-specific deletion of Piezo1 prevents the flow-induced vasoconstriction (revised Figure 6). Therefore the vasoconstriction is endothelium/flow-dependent.

“b) Additionally, repeating this experiment with Piezo1-knockout mesenteric vessels would mechanistically greatly corroborate the blood pressure findings and the hypothesis put forth by the authors.”

We have now done this as described above and we show the data in the revised Figure 6. The flow-induced vasoconstriction is endothelial Piezo1 dependent.

“2. The authors include figure 8 to demonstrate that the mesenteric resistance arterioles are unique in displaying Piezo1-dependent inhibition of endothelial-derived hyperpolarization. However, the comparison is made with major conductance arteries whose anatomy is starkly different from the 2nd order resistance arterioles tested in the mesentery. To actually make this point, the authors should strive to make this comparison between similar order arterioles from other organs, such as skeletal muscle.”

We used second-order mesenteric arteries, not arterioles. The typical diameter of these second-order arteries is shown in Figure 6. More precisely, the external diameter was $279.8 \pm 12.7 \mu\text{m}$ (n=17).

Regarding skeletal muscle arteries: The saphenous artery was chosen because it is almost identical in calibre to the second-order mesenteric artery: $284.2 \pm 13.4 \mu\text{m}$ (n=15). We added this information to the manuscript along with similar data for carotid artery.

“MINOR POINTS

Figure 7 raises more concerns than it actually answers, and in my opinion could be omitted from the manuscript. The explanation that the knockout mice work harder to on the treadmill is not substantiated by the ~2% change in bodyweight, which seems to be small enough to be well within the range of measurement error.”

Figure 7 importantly shows impact of the mechanism on whole body physical activity.

We are addressing a physiological mechanism and so not expecting large changes in body weight. A 2 % reduction in body weight is substantial in this context - especially within 7 days. If an 80 kg human lost 2 % of their weight in 7 days that would be 1.6 kg in 1 week, which is ~3.5 lb or a quarter of a stone.

The reduction in body weight was statistically significant in a blinded comparison of control and test groups in which the measurement error was the same for both groups. Therefore the difference cannot be explained by measurement error.

“In response to the authors rebuttal, the thallium-influx experiment indeed lacks the potentially confounding effect of release from intracellular stores, which a calcium assay has. However, thallium does cross essentially all potassium channels, of which there are plenty, and this assay can therefore be confounded by contribution to the observed signal from all manner of potassium channels. Additionally, calcium-dyes are much more sensitive. This is a minor point and does not take away from the value of the manuscript.”

It is correct that thallium ions go through potassium channels. This contributes a background signal which is inherent in this assay (and calcium assays). Importantly we show the Piezo1

component above background, both by over-expressing Piezo1 and by knocking down endogenous Piezo1 with siRNA.

REVIEWERS' COMMENTS:

Reviewer #1 (Remarks to the Author):

i am satisfied with the edits. This is now an excellent manuscript!